# peakTree: A framework for structure-preserving radar Doppler spectra analysis

Martin Radenz[1], Johannes Bühl[1], Patric Seifert[1], Hannes Griesche[1], and Ronny Engelmann[1]

[1]Leibniz Institute for Tropospheric Research (TROPOS), Leipzig, Germany

**Correspondence:** Martin Radenz (radenz@tropos.de)

**Abstract.** Clouds are frequently composed of more than one particle population even at smallest scales. Cloud radar observations frequently contain information on multiple particle species in the observation volume, when there are distinct peaks in the Doppler spectrum. Multi-peaked situations are not taken into account by established algorithms, which are only using moments of the Doppler spectrum. In this study, we propose a new algorithm, that recursively represents the subpeaks as nodes in a binary tree. Using this tree data structure to represent the peaks of a Doppler spectrum it is possible to drop all a-priori assumptions on the number and arrangement of subpeaks. The approach is rigid, unambiguous and can provide a basis for advanced analysis methods. The applicability is briefly demonstrated in two case studies, where the tree structure was used to investigate particle populations in Arctic multi-layered mixed-phase clouds, which were observed during the research vessel Polarstern expedition PS106 and the Atmospheric Radiation Measurement Programs' BAECC campaign.

## 1 Introduction

The characterization of mixed-phase clouds and associated microphysical processes poses a challenge to experimentalists, therefore these processes are still not well represented in general circulation models (Fan et al., 2011). In-situ instruments are subject to icing under the presence of supercooled liquid water, and the wide range of possible hydrometeor types require the deployment of instruments of which each can only cover a certain aspect of the whole hydrometeor distribution (Baumgardner et al., 2017; Korolev et al., 2017).

Cloud radars are frequently used for the investigation of mixed-phase clouds (Bühl et al., 2017). At Ka- and W-band, cloud radars are sensitive to scattering from the whole range of possible hydrometeors, ranging from cloud droplets to graupel (e.g. Kollias et al., 2007a; Fukao and Hamazu, 2014). In general, cloud radars are Doppler-capable and provide the backscattered signal as a function of Doppler velocity, commonly called Doppler spectrum (Wakasugi et al., 1986). When multiple particle populations are present in the observed volume, they are frequently represented as distinct peaks in the Doppler spectrum (e.g. Shupe et al., 2004; Luke et al., 2010; Verlinde et al., 2013; Yu et al., 2014; Kalesse et al., 2016; Kollias et al., 2016). The properties of a multi-peak situation can only partly be represented by the moments of a single peak algorithm, which causes errors in the target classification and subsequent microphysical retrievals. A multitude of approaches are available to classify clouds and retrieve water contents, particle sizes and number concentrations (for example Clothiaux et al. 2000; Wang and Sassen 2002; Wang et al. 2004; Hogan et al. 2006; Illingworth et al. 2007; an overview is provided in Shupe et al.

2016 and Zhao et al. 2012). Almost all established algorithms are based on the assumption of mono-modal hydrometeor size distributions, which likely causes significant errors in multi-peaked situations.

The analysis of multi-peaked Doppler spectra can be separated into three steps:

1. **peak identification** (or peak finding): locate the boundaries of a (sub-)peak

2. **peak structuring**: identify the arrangement of the (sub-)peaks

3. **peak interpretation**: categorize the peaks and interpret them

Most available methods focus either on the peak identification or the peak interpretation step. For peak identification either noise-floor separated peaks and/or local minima in the spectral reflectivity are used (Shupe et al., 2004; Rambukkange et al., 2011). More sophisticated approaches allow for a separation of multi-modal peaks. This is done for example by using skewness signatures (Luke and Kollias, 2013) or continuous wavelet transforms (Luke et al., 2010; Yu et al., 2014). Recently Kalesse et al. (2019) proposed an algorithm for subjective peak identification criteria using machine learning.

Structure is reflected by a linear list of all subpeaks, usually sorted by velocity or reflectivity. In a further step, Oue et al. (2018), using the microARSCL algorithm (Kollias et al., 2007b; Luke et al., 2008), allow a primary peak to be split into two subpeaks. But they constrain the structure by assuming the left peak (faster falling peak) to have a higher reflectivity. Additionally, a noise-floor separated secondary peak is possible, but this one is assumed to be mono-modal. Such strong constraints may be justified for short periods at single geographic locations, but are not suitable for a general approach. Up to now, no generic and flexible formalism is available to describe an arbitrary number of subpeaks of a Doppler spectrum without a-priori assumption on the structure.

In the peak interpretation step, the peaks are usually sorted into categories using their moments. Categories are for example one liquid and one ice peak (Shupe et al., 2004), liquid, newly formed ice, and ice from above (Rambukkange et al., 2011) or liquid and two ice populations (Oue et al., 2018).

This study will focus on the second step, peak structuring. It will be shown how a binary tree representation of multiple peaks can provide a rigid, hence flexible formalism for structuring the peaks in a Doppler spectrum. The tree structure allows an arbitrary number of subpeaks in any arrangement, while at the same time being unambiguous and easily accessible by algorithms. The software implementing the algorithm is easily applicable to other radar systems and available openly. The study is structured as follows: The datasets used for demonstration are introduced in Section 2. In Section 3, the binary tree peak structuring algorithm is presented. Section 4 is dedicated to the presentation of two case studies in which the algorithm was used to investigate Arctic mixed-phase clouds. Discussions and conclusions are covered in Section 5.

## 2 Datasets

The binary tree peak structuring algorithm is applied to two datasets using $K_a$ band radars from different manufacturers with slightly different settings. Details on the instruments and datasets are given here.

## 2.1 MIRA-35 during PASCAL

During the Physical feedbacks of Arctic planetary boundary level Sea ice, Cloud and AerosoL (PASCAL) campaign (PS106; Wendisch et al., 2019) a cloud radar MIRA-35 was operated as part of the OCEANET suite on R/V Polarstern together with, amongst other instruments, a Polly$^{XT}$ Raman and polarization lidar (Engelmann et al., 2016) and a HATPRO 14-channel microwave radiometer (Rose et al., 2005). MIRA-35 is a magnetron-based pulsed 35-GHz cloud radar with polarisation and Doppler capabilities (Görsdorf et al., 2015). During the campaign, MIRA-35 was operated in linear-depolarization-ratio (LDR) mode. The pulse repetition frequency was $5\,\mathrm{kHz}$ and one raw Doppler spectrum was based on the fast Fourier transform of $256$ pulses, yielding a Doppler resolution of $0.082\,\mathrm{m\,s^{-1}}$ (Tab. 1). The radar was operated in vertical pointing mode. It was based on a leveling platform which actively corrected for pitch and roll movement of the ship. Vertical movement of the radar was corrected at a rate of $4\,\mathrm{Hz}$ using the ship motion data originally recorded at $20\,\mathrm{Hz}$. For the datasets of Arctic clouds presented in here, the active stabilization was not available anymore due to a hardware failure. In the scope of this study, therefore the Doppler spectra acquired within $10\,\mathrm{s}$ were averaged incoherently to suppress the ship pitch and roll motion, while the vertical motion was still corrected at a rate of $4\,\mathrm{Hz}$. The lack of active pitch and roll suppression lead to an accuracy of the zenith pointing of $1.5°$. For horizontal wind velocities below $10\,\mathrm{m\,s^{-1}}$, the bias introduced to the observed vertical velocity thus is below $0.2\,\mathrm{m\,s^{-1}}$.

By default, MIRA-35 provides noise-cleaned compressed Doppler spectra (zspc) and moment data separately for meteorological targets and atmospheric plankton (Görsdorf et al., 2015). Further data analysis is subject to the operator of the cloud radar, to which the zspc data provides a solid base for potential application of peak separation techniques. Accurate measurements of polarization variables, like the LDR, depend strongly on instrument hardware due to polarization leakage. The lowest LDR observable (integrated cross-polarization ratio ICPR) with this version of MIRA-35 was estimated in the presence of light drizzle with the approach of Myagkov et al. (2015) and found to be $-27.6\,\mathrm{dB}$. A second effect that has to be considered while calculating the LDR, is the noise level in the cross channel. If the signal in the cross channel is below the noise level, the LDR is determined solely by the signal in the co channel and no meaningful information on the polarization state of the received signal can be derived (Matrosov and Kropfli, 1993). Hence, when calculating the LDR (Eq. A6) only bins where the signal in the cross channel is a factor of 3 above the noise level are taken into account.

## 2.2 KAZR during BAECC

The Biogenic Aerosols—Effects on Clouds and Climate (BAECC) campaign (Petäjä et al., 2016) was a deployment of the U.S. Department of Energy's Atmospheric Radiation Measurement (ARM) Mobile Facility (AMF) to Hyytiälä ($61.9°\mathrm{N}, 24.3°\mathrm{E}$), Finland from February to September 2014. A vertical looking KAZR $35\,\mathrm{GHz}$ cloud radar (Kollias et al., 2016) was part of the remote sensing instrumentation of the AMF. It was operated at a pulse repetition frequency of $2.8\,\mathrm{kHz}$. The Doppler resolution is $0.023\,\mathrm{m\,s^{-1}}$, as a 512 point fast Fourier transform was used to estimate the Doppler spectra (Tab. 1). The vertically pointing KAZR used in this campaign does not posses a cross channel, hence no polarimetric variables are available.

|  | PASCAL | BAECC |
|---|---|---|
| Type | MIRA-35 | KAZR |
| Frequency | $35\,\mathrm{GHz}$ | $35\,\mathrm{GHz}$ |
| Pulse Length | $200\,\mathrm{ns}$ | $333\,\mathrm{ns}$ |
| Integration Time | $10\,\mathrm{s}$ | $6\,\mathrm{s}$ |
| No. Incoherent Averages | 195 | 33 |
| Pulse Repetition Frequency | $5.0\,\mathrm{kHz}$ | $2.8\,\mathrm{kHz}$ |
| $N_{\mathrm{FFT}}$ | 256 | 512 |
| Nyquist velocity $v_{\mathrm{Nyq}}$ | $10.5\,\mathrm{m\,s}^{-1}$ | $5.9\,\mathrm{m\,s}^{-1}$ |
| Velocity resolution | $0.082\,\mathrm{m\,s}^{-1}$ | $0.023\,\mathrm{m\,s}^{-1}$ |

**Table 1.** Configuration settings of the two cloud radars used in this study.

## 3  Algorithm

### 3.1  Transforming the Doppler spectrum into the tree structure

The algorithm explained in here transforms each Doppler spectrum with its (sub-)peaks into a full binary tree structure. A full binary tree is a directed graph with one root node and recursively each node might possess either two child nodes or none (Garnier and Taylor, 2009). Each node can be identified by an index, which is based on level-order tree traversal. The index $i$ of a node can be calculated by the following formulas:

$$i_{\text{left child}} = 2\,i_{\text{parent}} + 1 \tag{1}$$

$$i_{\text{right child}} = 2\,i_{\text{parent}} + 2 \tag{2}$$

$$i_{\text{parent}} = \left\lfloor \frac{i_{\text{child}} - 1}{2} \right\rfloor \tag{3}$$

with the floor function $\lfloor\,\rfloor$. These indices are illustrated in Fig. 1. This example shows a complete binary tree, meaning all nodes are present. When applied, some indices, e.g. 9 and 10 or 5, 6, 11, 12, 13 and 14 might be missing. However, the rule 'a node might possess either two child nodes or none' always holds.

Applied to radar Doppler spectra, a node is related to a part of the Doppler spectrum that contains at least one peak. The peak boundaries are identified (step 1 as listed in Sec. 1) by a noise-floor threshold and local minima in the spectral reflectivity (or spectral power density). These boundaries are then used to construct the tree structure (step 2 as listed Sec. 1). The root node contains all signal of the Doppler spectrum above the noise threshold. Two example spectra from KAZR and MIRA-35 are shown in Fig. 2 (a) and (b), respectively. The boundaries and moments for the MIRA-35 and KAZR are listed in Tab. 2 and 3, respectively. In a first step, all noise-floor-separated peaks are added as child nodes with their boundaries $v_{\text{left}}$ and $v_{\text{right}}$ (in the example for MIRA-35 $-3.30$ and $1.32\,\mathrm{m\,s}^{-1}$). Each node is then checked for (sub)-peaks within using the peak boundaries from the lowest to the highest spectral reflectivity. Starting with the lowest minimum at $v_{\text{add}}$, the node containing this minimum

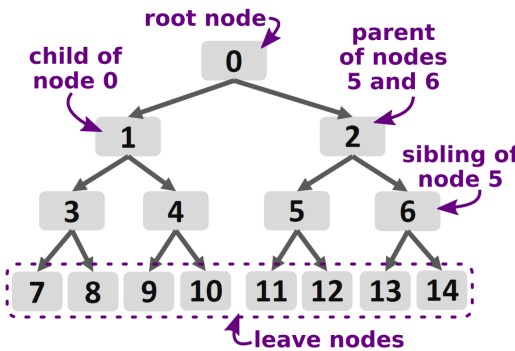

**Figure 1.** Binary tree containing 15 nodes with possible indices according to level-order tree traversal.

is split into two child nodes. When boundaries of the parent node are $[v_{\text{left}}, v_{\text{right}}]$, the left child node is $[v_{\text{left}}, v_{\text{add}}]$ and the right child node is $[v_{\text{add}}, v_{\text{right}}]$. In the example from Fig. 2 (b) the internal minimum with the lowest spectral reflectivity is at $-0.25\,\text{m s}^{-1}$ with a spectral reflectivity of $-33.4\,\text{dBZ}$. This reflectivity also defines the threshold, that separates the subpeaks. The splitting at local minima is repeated for all remaining minima, always splitting the leaf node (i.e. a node that does not have

5  any childs) in which the minimum is located.

A minimum is skipped, if the prominence of either of its subpeaks is less than $1\,\text{dB}$. Prominence is the difference between the maximum spectral reflectivity of a subpeak and the threshold that is defined as by the spectral reflectivity at local minimum (dashed grey lines in Fig 2 (a, b); similar to Shupe et al., 2004).

**Table 2.** Moments for each peak from the MIRA-35 Doppler spectrum depicted in Fig. 2 (b) with the index of the node according to the level-order tree traversal and the boundaries $v_{\text{left}}$, $v_{\text{right}}$ are given in $\text{m s}^{-1}$. Child-nodes are denoted by their level of indentation. The units are dBZ for reflectivity $Z$ and $\text{m s}^{-1}$ for $\overline{v}$ and spectral width $\sigma$. The skewness $\gamma$ is dimensionless, LDR is in dB. The threshold 'thres.' is in dBZ and the prominence 'prom.' is in dB.

| Node index | Boundaries $[v_{\text{left}}, v_{\text{right}}]$ | $Z$ | $\overline{v}$ | $\sigma$ | $\gamma$ | LDR | thres. | prom. |
|---|---|---|---|---|---|---|---|---|
| 0 | `‘−` [-3.30, 1.32] | -11.57 | -1.10 | 0.59 | 1.01 | -25.3 | -52.1 | 32.0 |
| 1 | `+−` [-3.30, -0.25] | -12.19 | -1.27 | 0.36 | 1.08 | -26.9 | -33.4 | 13.2 |
| 3 | `\|` `+−` [-3.30, -1.07] | -13.27 | -1.44 | 0.15 | 0.27 | -26.1 | -28.7 | 8.5 |
| 4 | `\|` `‘−` [-1.07, -0.25] | -18.35 | -0.81 | 0.16 | -0.13 | -32.2 | -28.7 | 1.4 |
| 2 | `‘−` [-0.25, 1.32] | -20.08 | 0.04 | 0.13 | -0.31 | -20.9 | -33.4 | 6.2 |

In the next step, the moments of the Doppler spectrum (reflectivity, mean velocity, width, skewness) are calculated for
10  each node within its boundaries $[v_{\text{left}}, v_{\text{right}}]$ (see Appendix A). Equivalent reflectivity factor $Z$ (the subscript $e$ is omitted for brevity) is calculated by integrating the spectral reflectivity of the whole peak (i.e. from the noise-floor up). For all higher moments, signal below the threshold, that separated the (sub-)peak is neglected to avoid biases (see also Fig. A1). The LDR for each node is calculated using the spectral reflectivity in the cross channel, if such a channel is available.

# Dopplerspectrum

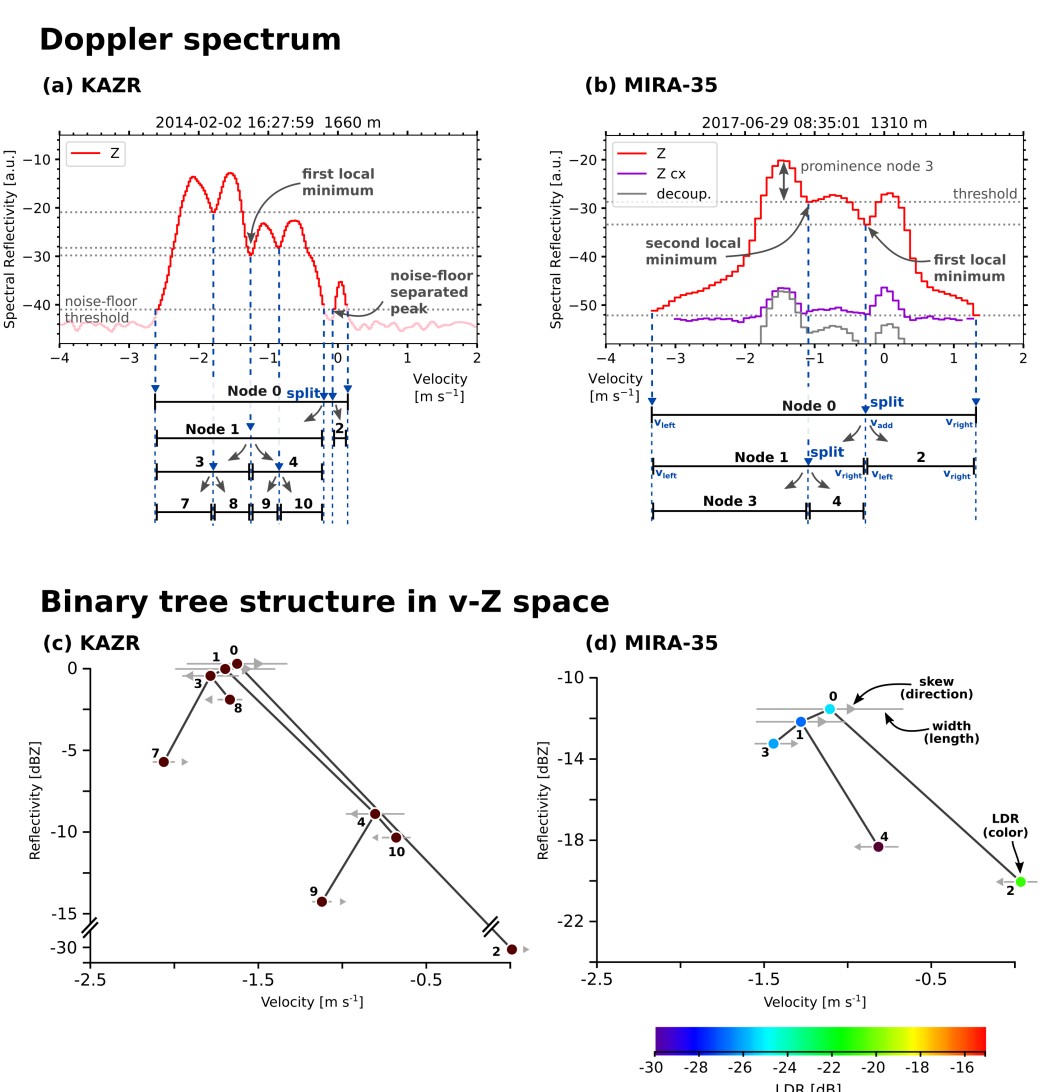

**Figure 2.** Example for generating the trees for two Doppler spectra from different cloud radars of type KAZR (a, c) and MIRA-35 (b, d). The root node (Node 0) is splitted into child-nodes at the indicated velocity bins (dashed blue) which contain a local minimum in spectral reflectivity. The threshold defined by the noise-floor and the internal minima is marked with dashed grey lines. In the spectrum in (b) the reflectivity in the cross channel (Z cx) is shown together with the co channel signal subtracted by the polarization decoupling (further explanation in Sec. 2.1). (c) and (d) show the resulting trees, where the location of a node in the v-Z space is based on its moments. Spectral width is indicated quantitatively by the length of the grey lines and sign of the skewness is indicated by a triangle (pointing to left for negative skewness and vice-versa). The circle denoting the nodes position is color-coded in accordance to the nodes LDR.

Node 0 contains all components of the Doppler spectrum which are above the noise threshold. In general, this node is similar to the moment estimation commonly used to analyze Doppler spectra (e.g. Carter et al., 1995; Clothiaux et al., 2000;

**Table 3.** Moments for each peak from the KAZR Doppler spectrum depicted in Fig. 2 (a) similar to Tab. 2

| Node index | Boundaries $[v_{\text{left}}, v_{\text{right}}]$ | $Z$ | $\overline{v}$ | $\sigma$ | $\gamma$ | thres. | prom. |
|---|---|---|---|---|---|---|---|
| 0 | `− [-2.58, 0.12] | 0.78 | -1.69 | 0.41 | 1.04 | -41.0 | 28.2 |
| 1 | +− [-2.58, -0.21] | 0.78 | -1.70 | 0.41 | 0.99 | -41.0 | 28.2 |
| 3 | \| +− [-2.58, -1.26] | 0.31 | -1.80 | 0.27 | -0.12 | -29.8 | 17.0 |
| 7 | \| \| +− [-2.58, -1.78] | -2.73 | -2.04 | 0.11 | 0.28 | -20.9 | 7.3 |
| 8 | \| \| `− [-1.78, -1.26] | -2.61 | -1.56 | 0.09 | -0.25 | -20.9 | 8.1 |
| 4 | \| `− [-1.26, -0.21] | -9.09 | -0.81 | 0.23 | -0.17 | -29.8 | 7.2 |
| 9 | \| \| +− [-1.26, -0.85] | -12.84 | -1.04 | 0.09 | 0.25 | -28.2 | 5.1 |
| 10 | \| \| `− [-0.85, -0.21] | -11.38 | -0.65 | 0.10 | -0.18 | -28.2 | 5.6 |
| 2 | `− [-0.05, 0.12] | -28.06 | 0.04 | 0.04 | -0.03 | -41.0 | 5.7 |

Görsdorf et al., 2015). Only in case of the presence of noise-separated subpeaks within node 0, some moment estimators such as microARSCL apply the moment retrieval to the most significant peak only. The child nodes (1 and 2) of node 0 are the subpeaks defined by the lowest relative minimum. The second lowest minimum then splits one of these nodes and gives nodes 3 and 4 (splitting node 1) or 5 and 6 (splitting node 2). The total number of subpeaks $n_{\text{subpeaks}}$, as estimated by established peak finding methods, can be calculated from the number of nodes $n_{\text{nodes}}$:

$$n_{\text{subpeaks}} = (n_{\text{nodes}} + 1)/2 \tag{4}$$

Each node is characterized by its reflectivity $Z$, vertical velocity $v$, spectral width, skewness, LDR and prominence. It is suitable to visualize the tree in the v-Z plane as a color-filled circle with the parent-child relationships depicted by a black line (Fig. 2 c and d) and each circle is color-coded in accordance to its LDR (if available as for MIRA-35). The width and skewness are shown by a horizontal grey line and a grey triangle with varying size, respectively. This representation hence combines all key parameters of a multipeak Doppler spectrum.

## 3.2 Peak interpretation

Once the tree structure is constructed various methods for peak interpretation can be used. In this study only two rather basic approaches are shown.

### 3.2.1 Selecting cloud droplet nodes

Nodes, that are most likely caused by liquid droplets can be identified by their moments, as done already in previous studies (e.g. Frisch et al., 1995; Shupe et al., 2001; Rambukkange et al., 2011; Yu et al., 2014; Kalesse et al., 2016). The liquid cloud droplets are assumed to be small and only possess a negligible terminal velocity. In the absence of strong vertical air motions the (sub-)peak caused by the liquid droplets will be close to $0\,\text{m}\,\text{s}^{-1}$. Additionally, due to their small size, the reflectivity of these droplets is rather small. Combining this to characteristics, a simple selection rule is based on two thresholds: $Z < -20\,\text{dBZ}$

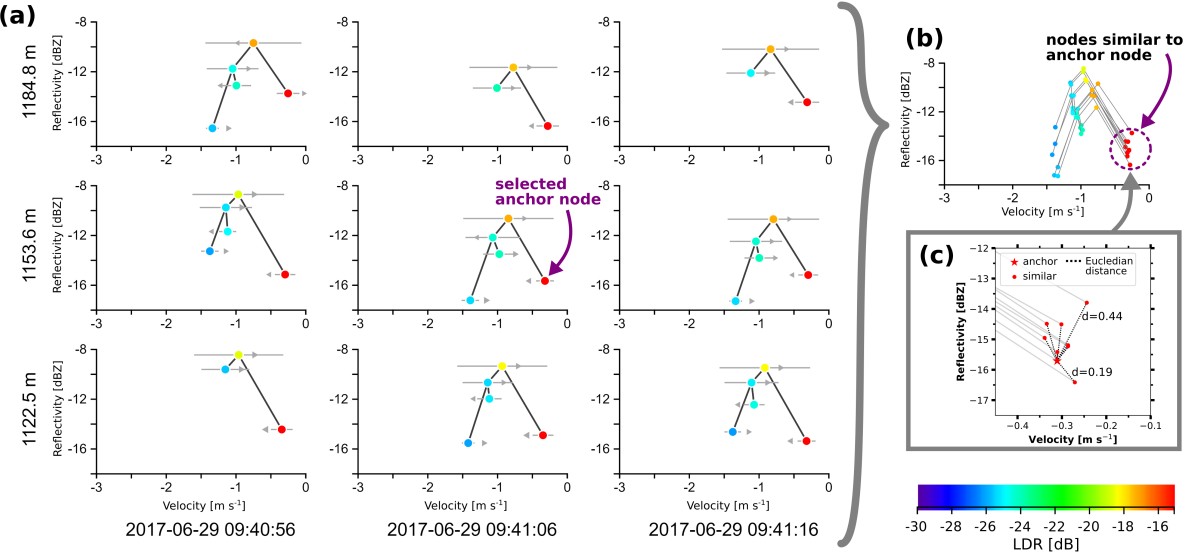

**Figure 3.** Illustration of the grouping algorithm for one anchor node. Single trees for a time-height slice of MIRA-35 observations are depicted in (a) with the selected anchor node marked by an arrow. The time-height cross section shown omits the outer trees of the $5 \times 5$ slice for clarity. The moments of each node are illustrated, as described in Fig. 2. In (b), the trees are combined into the same v-Z illustration with a circle denoting the nodes that are identified as similar by the Euclidean distance criterion. The Euclidean distance $d$ is depicted in inset (c) for all nodes with index 2.

and $|v| < 0.3 \, \mathrm{m \, s^{-1}}$. Using this criteria, each node in a tree can be checked and the index of the fitting node is stored. This selection process can be used on larger time-height slices straightforward and computationally efficient, identifying regions of a cloud, where liquid droplets cause a distinguishable (sub-)peak.

### 3.2.2 Grouping nodes into particle populations

The nodes representing similar particle populations in neighboring (time-height) bins can be connected to obtain a continuous picture of the evolution of a particle population. First, a node is manually assigned to a particle population based on visual inspection and guided by the LDR value. These manually selected (anchor) nodes are spaced in steps of $50 \, \mathrm{s}$ and $150 \, \mathrm{m}$, making one anchor representative for a slice of 5 time steps and 5 height bins or, in other words, for the 25 neighboring trees. For the time-height bins in between these anchor nodes, nodes with similar characteristics of the moments are automatically selected. Similarity is given, if a node is close to the anchor node in the v-Z space in minimal in terms of Euclidean distance $d$ and below a threshold $d < 0.9$. For the present study, the parameters $Z$ and $v$ are normalized by factors of 5 and 0.3 respectively, to make both comparable for the grouping algorithm (Fig. 3). The sibling of each selected node is afterwards assigned to the complementary particle population.

## 4 Application

### 4.1 KAZR case, 02 February 2014: Identifying embedded liquid layers

On the 02 February 2014 an intense low pressure system over Faroe Islands together with a high pressure core above South-Western Russia caused southerly flow over Finland. Low level clouds were observed the whole day. During the afternoon a occluded front with cold frontal character reached the Hyytiälä field site (further description also in Kalesse et al., 2019). Frontal precipitation started at 14:30 UTC and was caused by two geometrically deep cloud systems, both topped at around $8\,\mathrm{km}$ height with a short pause in between. The first precipitation event was characterized by liquid precipitation with a melting layer at around $1.2\,\mathrm{km}$ height (not shown), but for the second event from around 15:30 UTC onwards snow precipitation dominated. Fig. 4 shows a part of the frontal system. Between 16:00 and 16:20 UTC a lower cloud with almost constant cloud top at $2.6\,\mathrm{km}$ height was observed. In this cloud, reflectivity and vertical velocity are increasing toward the ground. At 16:18 UTC ice crystals from the top cloud start to sediment into this lower cloud. At $2.2\,\mathrm{km}$ height, the vertical velocity of these particles increases up to $-2.0\,\mathrm{m\,s^{-1}}$ (Fig. 4 b). The total number of nodes (Fig 4 c) reveals, that the cloud radar Doppler spectra were dominated by multi-peaked situations, predominantly in the lower layer. Up to 9 nodes were found, which corresponds to 5 subpeaks.

The selection rule described in Sec. 3.2.1 is now used to identify nodes, that are likely caused by liquid droplets. Fig. 5 shows the moments of the liquid droplet nodes. Two liquid layers become visible, a lower one between $0.8\,\mathrm{km}$ and $1.3\,\mathrm{km}$ height and a higher one with more irregular boundaries between $1.6\,\mathrm{km}$ and $2.6\,\mathrm{km}$ height. The vertical velocity (Fig 5) shows the typical pattern of small particles with negligible terminal fall velocity, which follow the air motion. Areas of up- and downdrafts with velocities between $-0.5\,\mathrm{m\,s^{-1}}$ and $+0.5\,\mathrm{m\,s^{-1}}$ are clearly visible.

### 4.2 MIRA-35 case, 29 June 2017: Separating two ice crystal populations in an Arctic cloud

On the 29 June 2017 R/V Polarstern was located a few nautical miles north of the island Kvitøya at $80.5°\mathrm{N}, 31.5°\mathrm{E}$ and operated in the frame of the PASCAL campaign. The synoptic situation was controlled by a low over Fram strait with a secondary low that passed Polarstern on that day with the surface wind veering from SE to NW and frequent light precipitation.

Between 08:30 and 09:45 UTC a cloud was continuously observed by MIRA-35 from the surface up to $2.7\,\mathrm{km}$ height with a cloud top temperature of $-15°\mathrm{C}$ (see Fig. 6). The thermodynamic structure of the cloud was probed by a RS92-SGP radiosonde, that was launched from Polarstern at 10:50 UTC (Schmithüsen, 2017). The spread between temperature and dewpoint (Fig. 6 a) shows saturation with respect to liquid water throughout the whole cloud. Very light precipitation was observed at the surface by an optical disdrometer (Klepp et al., 2018), peaking to $0.1\,\mathrm{mm\,h^{-1}}$ at 08:50 UTC. The highest values of liquid water path ($\sim 50\,\mathrm{g\,m^{-2}}$), obtained from the microwave radiometer (Rose et al., 2005), were also observed during this time. Low reflectivity and vertical velocities close to $0\,\mathrm{m\,s^{-1}}$ with alternating up- and downdrafts suggest the presence of a turbulent liquid layer capping the cloud. Below $1.3\,\mathrm{km}$ height, reflectivity and LDR of the single peak analysis show a sharp increase, giving hints to a change in microphysical properties, such as size or shape.

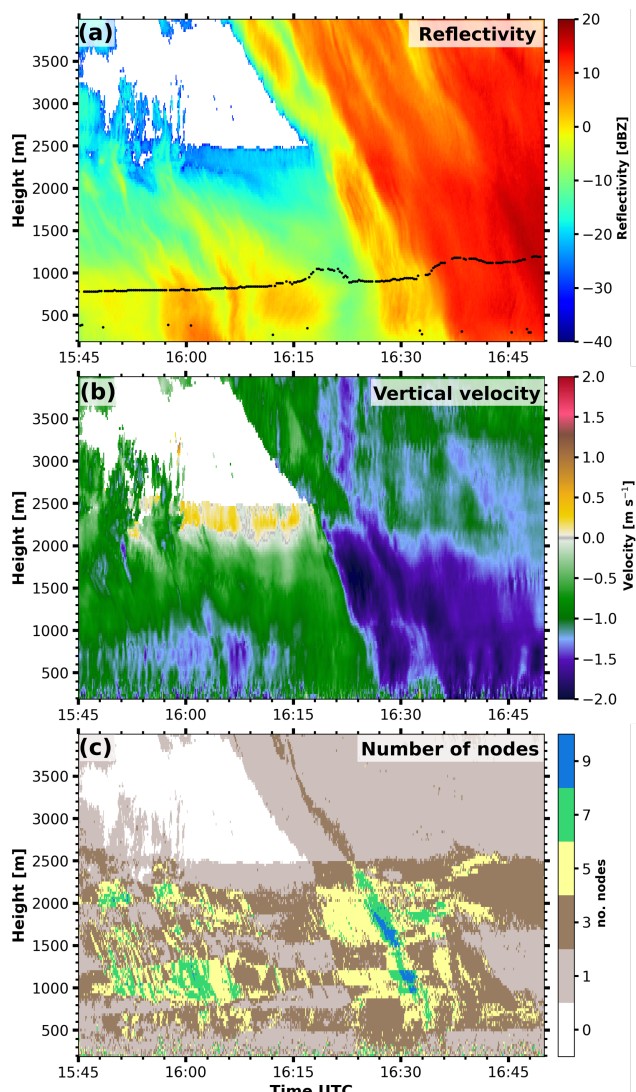

**Figure 4.** KAZR reflectivity with lidar detected cloud base height (black dots, a), mean velocity (b) and number of nodes (c) at the 02 February 2014 from 15:45 to 16:45 UTC.

Application of the multi-peak analysis introduced above reveals that multi-peak spectra were quite frequent (Fig. 6 f). Fig 7 shows the nodes identified as caused by liquid droplets according to the selection rule from Sec. 3.2.1. Two continuous liquid layers at almost constant heights were observed during the whole event. The uppermost layer at $2.7\,\mathrm{km}$ height topping the cloud is also visible in the moments of the full spectrum. The lower layer at $1.3\,\mathrm{km}$ height, being hidden when only using the moments of the full spectrum. Furthermore, shorter periods with likely liquid water presence were detected, for example from 08:55 to 09:15 UTC at $1.0\,\mathrm{km}$ height. Additionally, the ceilometer-detected cloud base (Fig 7 a, black dots) indicates the

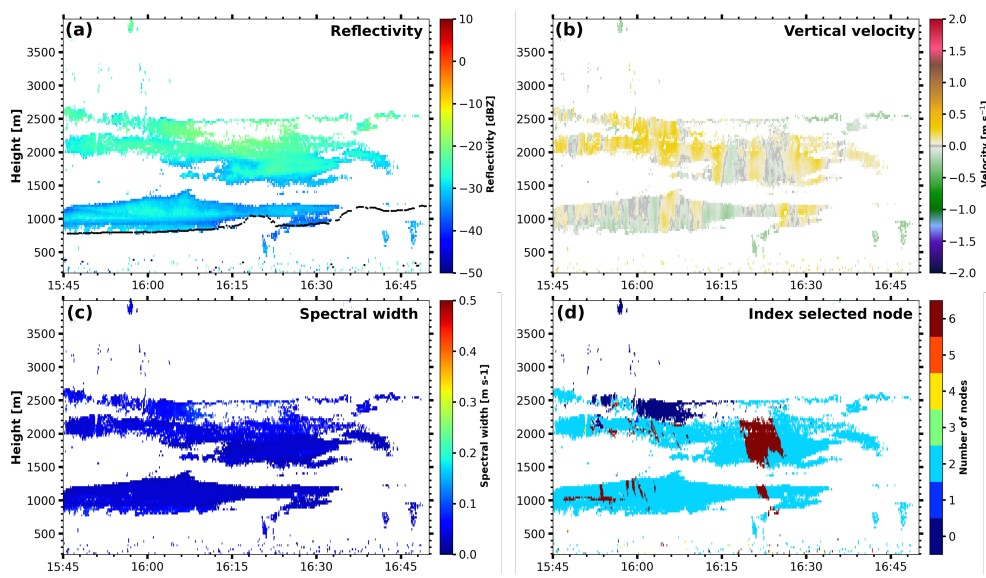

**Figure 5.** KAZR reflectivity with lidar detected cloud base height (black dots, a), mean velocity (b), spectral width (c) and index of node (d) for nodes identified as liquid cloud drops at the 02 February 2014 from 15:45 to 16:45 UTC.

base of the liquid layer at $750\,\mathrm{m}$ height between 08:40 and 09:15 UTC. This lower part of the liquid layer did not produce a distinguishable (sub-)peak in the Doppler spectrum and therefore no individual node.

After grouping the nodes to particle populations (as explained in Section 3.2.2), the microphysical structure of this cloud becomes clearer. The faster-falling particle population (Fig. 8 left column) originating at the uppermost liquid layer at $2.7\,\mathrm{km}$
height has a rather variable reflectivity with background values of around $-20\,\mathrm{dBZ}$ and maxima in frequently occuring fall-streaks of up to $0\,\mathrm{dBZ}$ reflectivity. The vertical velocity (Fig. 8 c) is quite variable as well. Below $2.5\,\mathrm{km}$ height, the ice particles generated at cloud top descend with velocities of $-0.5$ and $-2.0\,\mathrm{m\,s^{-1}}$. The low LDR of these particles (Fig. 8 e) is characteristic for oblate or plate-like particles (Myagkov et al., 2016), which is also consistent with particle shapes formed at cloud top temperatures of around $-15°\mathrm{C}$ (Bühl et al., 2016). Below the height of primary ice formation, several processes like
depositional growth and and aggregation might contribute to particle growth.

Frequently, fallstreaks from the upper layer penetrate the second liquid layer at $1.3\,\mathrm{km}$ height. The lower-level liquid layer with a temperature of $-5°\mathrm{C}$ also continuously forms ice (Fig. 8 right column). The vertical velocity (Fig. 8 d) is lower ($-0.2\,\mathrm{to}-0.7\,\mathrm{m\,s^{-1}}$) and more homogeneous than for the other particle population. The high LDR of $-14\,\mathrm{dB}$ at heights of $100$ to $200\,\mathrm{m}$ below the top of the liquid layer can be attributed to prolate or columnar ice crystals (Myagkov et al., 2016;
Bühl et al., 2016). Hence, ice formation takes place between $1.1$ and $1.3\,\mathrm{km}$ height, which is also underpinned by the gradual increase of reflectivity and vertical velocity toward ground. Below $1.1\,\mathrm{km}$ height, the reflectivity is more variable, with maxima of $9\,\mathrm{dBZ}$ at 08:50 UTC and minima of $-11\,\mathrm{dBZ}$ at around 09:10 UTC. We can not fully rule out from the information given, that ice multiplication was triggered when the higher-level ice particles descended into the lower liquid layer. However, ice was

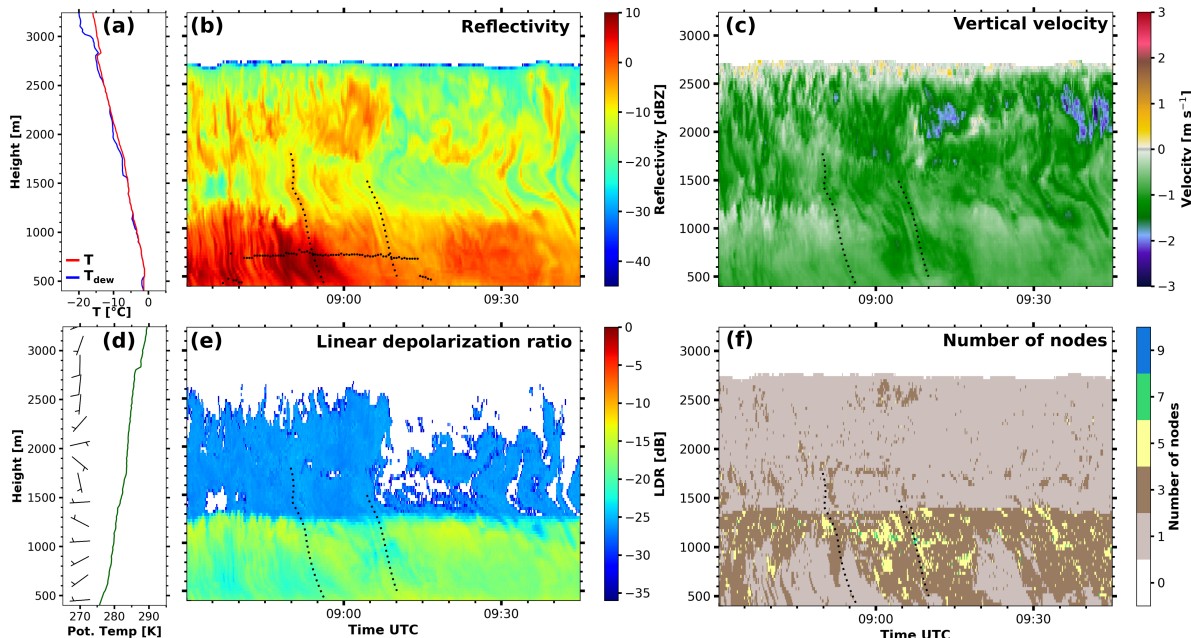

**Figure 6.** Radiosonde ascend at the 29 June 2017 10:50 UTC (a, d from Schmithüsen, 2017). MIRA-35 reflectivity with lidar detected cloud base height (black dots, b), mean velocity (c) and linear depolarization ratio (e) of the zeroth node (moments of the single peak analysis) at the 29 June 2017 from 08:30 to 09:45 UTC. Total number of nodes (f) for the same period.

formed from the lower liquid layer constantly over time (Fig. 8 right column), even in periods where particles with very low reflectivity were potentially seeding from above, as it was the case for example between 09:10 und 09:30 UTC. This supports the interpretation, that at least a few ice crystals were caused by primary ice formation.

Looking into two individual fallstreaks, it is possible to track the evolution of the two particle populations. The selected
5  fallstreaks are illustrated as black dashed curves in Fig. 8. In the frame of this study, the fallstreaks were tracked manually based on the criterium of following the maximum of the radar reflectivity of the faster falling (and also oblate) particle population. It should be noted, that techniques for an automated classification of fallstreaks exist (Kalesse et al., 2016; Pfitzenmaier et al., 2017), which should be applied when longer time series are analysed. But these algorithms have shortcomings when strong directional wind shear is present, as in this case (Fig. 6 d). First hints for different microphysical processes become evident from
10  tracking the properties of the individual nodes in the selected fallstreaks. The first one starts at 08:52 UTC and $1.8\,\mathrm{km}$ height with oblate particles having rather constant reflectivity of $-5\,\mathrm{dBZ}$ and a vertical velocity of around $-1.0\,\mathrm{m\,s^{-1}}$. The reflectivity of this population is almost constant down to $0.9\,\mathrm{km}$ height, even after the fallstreak reaches the lower liquid layer. The LDR is unaffected by the liquid layer as well. Contrarily, the prolate particle population generated within this liquid layer shows a strong increase in reflectivity from $-20\,\mathrm{dBZ}$ to $+6\,\mathrm{dBZ}$, while LDR decreases from $-14\,\mathrm{dB}$ to $-19\,\mathrm{dB}$. Below $0.8\,\mathrm{km}$ height,

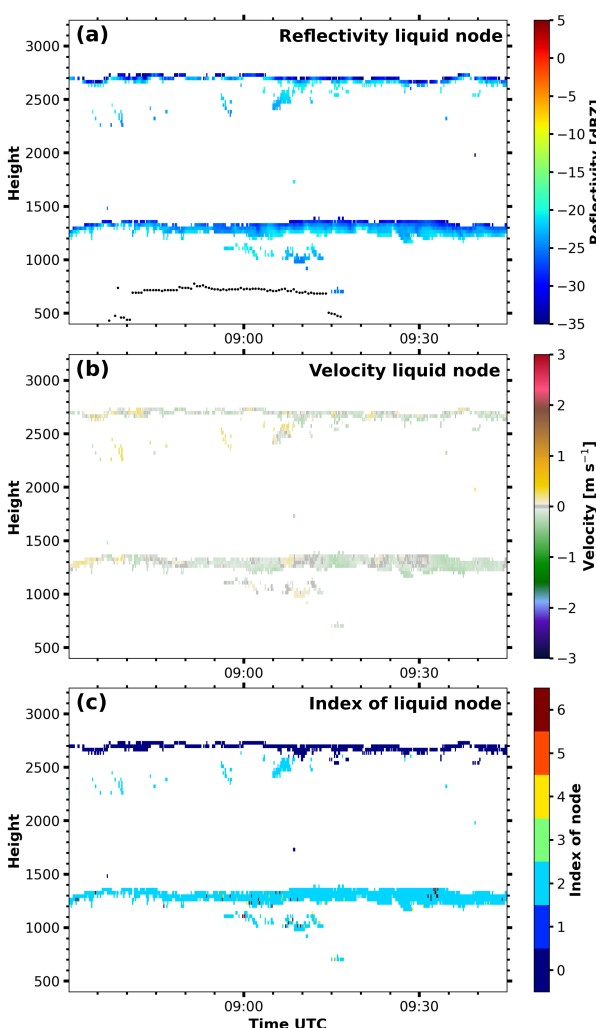

**Figure 7.** MIRA-35 reflectivity with lidar detected cloud base height (black dots, a), mean velocity (b) and index of node (c) for nodes identified as liquid cloud drops by the selection rule from Sec. 3.2.1 at the 29 June 2017 from 08:30 to 09:45 UTC.

the faster falling mode (low LDR) is not longer visible as a separate peak (and accordingly the node disappears), because the slower falling population (one with high LDR) becomes dominant in the Doppler spectrum.

The second fallstreak, being less pronounced than the first one, begins at 09:06 UTC and $1.5\,\mathrm{km}$ height with a reflectivity of $-10\,\mathrm{dBZ}$ and again a vertical velocity of around $-1.0\,\mathrm{m\,s^{-1}}$ for the oblate particles. After reaching the liquid layer at 5 $1.3\,\mathrm{km}$ height, the reflectivity of this particle population increases to $-7\,\mathrm{dBZ}$ and also the velocity increases slightly. The LDR remains below $-25\,\mathrm{dB}$. The second particle population with a prolate shape grows as well. From less than $-20\,\mathrm{dBZ}$ in the liquid layer to $-4\,\mathrm{dBZ}$ at $0.6\,\mathrm{km}$ with a final velocity of $-0.6\,\mathrm{m\,s^{-1}}$. During this growth, LDR remains at the high value of $-14\,\mathrm{dB}$, indicating no change of the particle shape. Due to insufficient polarimetric data, especially the lack of scans, it is

difficult to disentangle the contribution of different microphysical processes to particle growth. A more detailed investigation, using synergistic retrievals on top of the algorithm presented here, is required to pin-down the relevant processes further.

The ice water content (IWC) for each particle population can be retrieved from $Z$ and the temperature (Hogan et al., 2006). This $Z - T$ retrieval was developed under the assumption of mono-modal peaks in the Doppler spectrum, but using the tree

structure it is possible to include the information from the Doppler spectrum into this retrieval rather easily. Fig. 9 shows the IWC for node 0 or the full Doppler spectrum (a), hence assuming single-peaked spectra. Applying the IWC retrieval for the separated particle populations reveals the relative contribution of one population to total ice mass (Fig. 9 b). As could also be seen in the discussion on the reflectivity of the particle populations above, the precipitation reaching the ground between 08:30 and 09:00 UTC could not be directly linked to cloud top ($2.7\,\mathrm{km}$), as the whole ice mass below $1.3\,\mathrm{km}$ height can be assigned

to the prolate particle population. Contrarily between 09:00 and 09:15 ice mass was almost equally distributed to both particle populations below $1.3\,\mathrm{km}$ height. Following this approach, the proposed technique can also be used to extend the capabilities of other established retrieval algorithms.

## 5   Discussion and Conclusions

We proposed a binary tree structure for individual peaks of a multi-peaked cloud radar Doppler spectrum. This data structure

does not require prior assumptions on the arrangement or hierarchy of the peaks. The tree structure allows to select the level of complexity with which the Doppler spectrum is approximated, by the number of nodes taken into account. It also provides backward compatibility, as the root node (i.e. node 0) holds the moments of the full Doppler spectrum with an implicit assumption of mono-modality. These moments are similar to standard Doppler spectra processing. Hence, a seamless transition from current single-peak techniques to multi-peak analysis is possible. The recursive structure of the tree allows to drop the

artificial separation into noise-floor separated peaks and subpeaks within noise-floor separated peaks, as was necessary in prior approaches (see Fig 13 in Williams et al., 2018). This separation imposed strong constraints on structure, without having a physical reason. For example, depending on the intensity of turbulence, two noise-floor separated peaks might appear only as (sub-)peaks of one peak in the Doppler spectrum.

We showed two basic techniques to demonstrate, how the tree structure can be utilized for peak interpretation (step 3 as

defined in Sec. 1). The first technique used a simple selection rule to identify peaks that most likely are caused by liquid water. The choice of the threshold is based on prior studies. For reflectivity, the $-20\,\mathrm{dBZ}$ threshold as used by Oue et al. (2018) or reported by Kalesse et al. (2016), is rather conservative compared to older studies (e.g. Frisch et al., 1995; Kogan et al., 2005; Yu et al., 2014). The velocity threshold depends strongly on dynamical environment. The threshold of $|v| < 0.3\,\mathrm{m\,s^{-1}}$ might only be valid for stratiform conditions, as in the two case studies shown. Good agreement with the ceilometer-derived

cloud base heights in the KAZR case (Sec. 4.1) is in agreement for other studies using comparable threshold. Similar rules can potentially be applied for other particle types with a clear signature in the moments of the Doppler spectrum, as it might be for example be the case for heavily rimed particles.

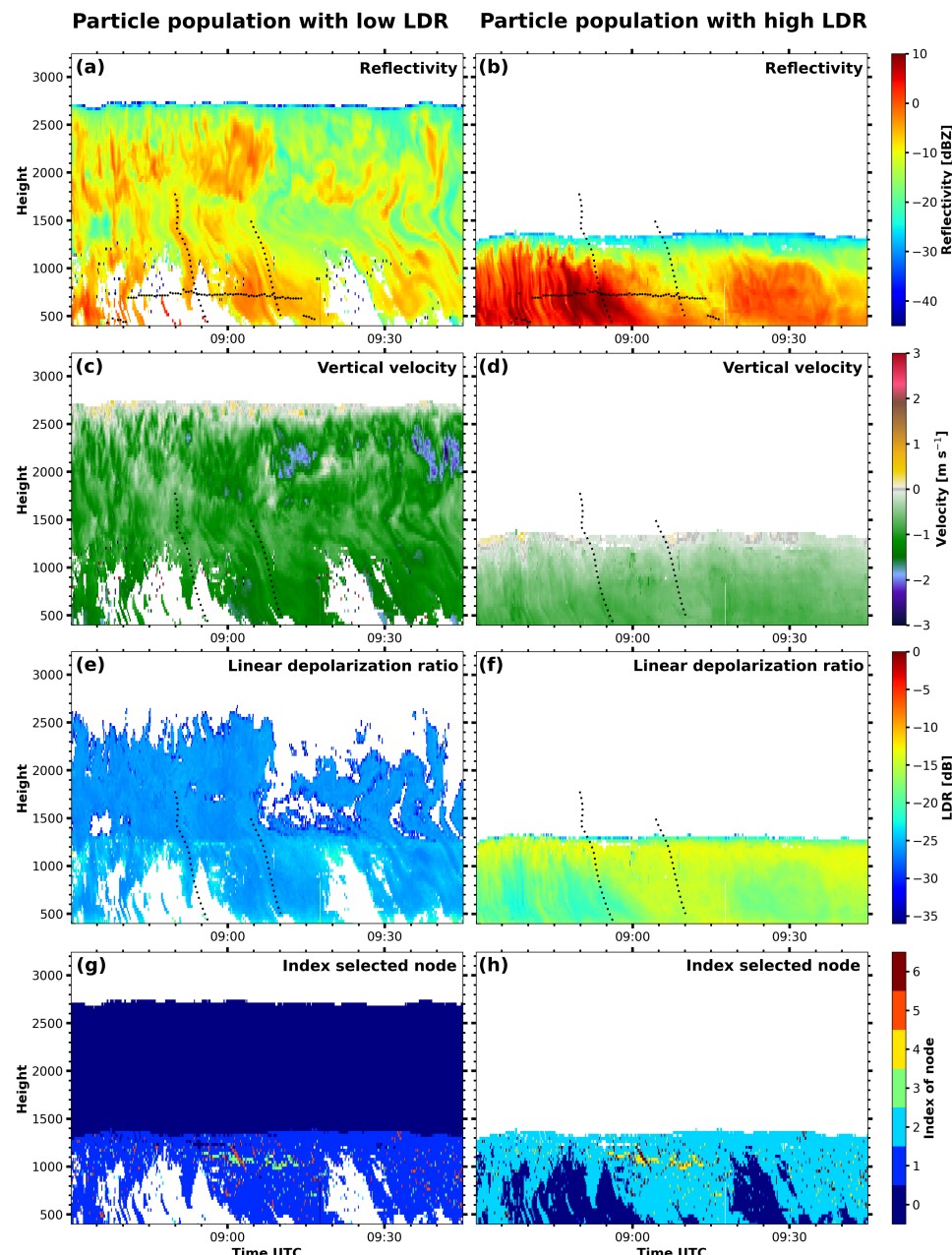

**Figure 8.** MIRA-35 reflectivity with lidar detected cloud base height (black dots, a, b), mean velocity (c, d), linear depolarization ratio (LDR; e, f) and index of the selected node (g, h) of the two particle populations (left and right column) at the 29 June 2017 from 08:30 to 09:45 UTC. The dashed black lines locate the two fallstreaks described in the text. For regions marked in white no node could be assigned to the respective particle population.

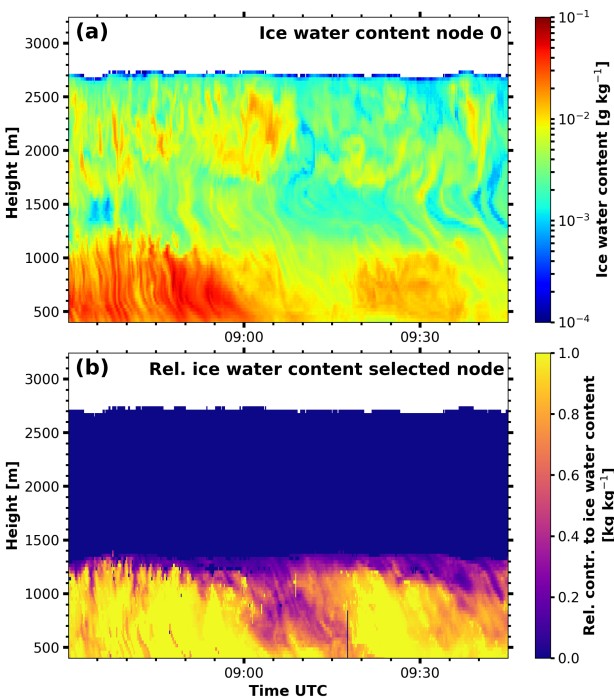

**Figure 9.** Retrieved ice water content from node 0 (a) and the relative contribution of the particle population with high LDR (b) at the 29 June 2017 from 08:30 to 09:45 UTC.

The second technique grouped nodes from neighboring Doppler spectra into particle populations based on their moments using the Euclidean distance in v-Z space. The thresholds used here depend strongly on the conditions they are applied to. The normalization factors for reflectivity $Z$ and velocity $v$ weight the variation in one dimension with respect to the other. For this case, changes in velocity were only allowed to be relatively small compared to changes in reflectivity. Considering only changes along one axis, reflectivity could vary by $4.5\,\mathrm{dBZ}$ for neighboring trees for two nodes still be considered to belong to the same particle population, whereas velocity could only vary by $0.27\,\mathrm{m\,s^{-1}}$. The distance threshold $d$ controls gaps in the grouping. A low distance threshold would select only a subset of nodes, introducing gaps in time and range for the grouped particle population. A high distance threshold can make the selection ambiguous, as two or more nodes in one tree might fulfill this condition. The frequency of anchor nodes is controlled by the evolution of the cloud. When rapid changes are expected, as for example in more convective situations, more anchor nodes will be required. In this study, the anchor nodes had to be selected manually, but automatizing this selection should be also possible in a future step.

The binary tree peak structuring algorithm together with the interpretation techniques was applied to mixed-phase cloud cases observed by two different cloud radar systems. The liquid node selection rule (Sec. 3.2.1) was applied to to case studies from both campaigns. In the KAZR case study from the BAECC campaign multi-peaked Doppler spectra were quite frequent (Sec. 4.1). Selecting the nodes that were likely caused by liquid droplets revealed two liquid layers. The base of the lower layer

at $800\,\text{m}$ was also observed by a collocated ceilometer. In the MIRA case (Sec. 4.2), the liquid droplets detected at cloud base at around $750\,\text{m}$ height did not form a distinct peak and can not be identified by this basic approach. More sophisticated peak identification methods (step 1 as defined in Sec. 1) could be used to address this issue.

The grouping technique (Sec. 3.2.2) was demonstrated in a second case of an Arctic mixed-phase cloud, where nodes were separated and sorted into two particle populations. Looking at both particle populations individually provides deeper insights into the prevalent physical processes. The upper liquid layer formed ice particles of, most likely, oblate shape as indicated by the LDR. While sedimenting, these particles grew further, either due to water vapor deposition or aggregation. When reaching a second liquid layer below, riming becomes available as a potential third growth process. Within this liquid layer, a new ice particle population emerges. Using the tree representation of multi-peaked Doppler spectra, we were able to identify this second liquid layer and individually track the evolution of the upper-level and the new ice particle population. Indications are given that new particles are formed: (1) the LDR signatures point toward prolate particles, which fits to the temperature in the second liquid layer and (2) ice is also produced in regions where ice fallstreaks from above are absent.

Nevertheless, the characterization of the interactions between the two populations and further narrowing-down relevant growth processes would require a more-detailed investigation based on polarimetry or multi-wavelength radar and lidar synergy, which is beyond the scope of the current study. Furthermore, this case study covers situations, where the assumption of the fastest falling subpeak was not the one with the highest reflectivity, as done by Oue et al. (2018), was violated.

In summary, we consider the binary tree peak structuring algorithm a well-suited approach for enhancing the capabilities of cloud radars for the analysis of multi-peaked spectra, especially the characterization of mixed-phase cloud processes. Tracing the evolution of polarimetric properties and velocity of distinct nodes will allow much more detailed studies of the ice growth and ice multiplication processes in future.

It is feasible to apply the algorithm also to Doppler spectra of further radars, as only very few parameters, namely the number of incoherent averages, the prominence, the noise threshold and - if a cross channel is available - the ICPR, need to be adjusted.

This study used rather basic techniques for peak identification and interpretation (steps 1 and 3 as listed in Sec. 1). Both issues can be addressed in further work, but keeping the tree structure as a connection. For example other peak finding algorithms can easily be used to replace the minimum in spectral reflectivity approach used here. The only information required to build the tree are at which Doppler bins the Doppler spectrum should be splitted. With respect to the interpretation step, the tree structure can extend the capabilities of established classification algorithms and microphysical retrievals. Many of these methods are based on single moment data and hence a mono-modal assumption. By applying the retrieval to each node individually, the strong assumption of mono-modality could be relaxed without major adjustments in the retrieval algorithm itself, as shown for the $Z - T$ ice water content retrieval.

*Code and data availability.* The processing software "peakTree" as used for this publication is available under Radenz et al. (2019). The most recent version is available via GitHub: https://github.com/martin-rdz/peakTree (last access: 25.02.2019). The radiosonde data is available by

Schmithüsen (2017) and the cloud radar Doppler spectra are available on request. The KAZR data for the BAECC case study is available from the ARM data center under http://www.archive.arm.gov (Bharadwaj et al., 2014).

## Appendix A: Formulas for calculating the moments

The moments for each node in the Doppler spectrum are calculated following the formulas given by Maahn and Löhnert (2017), Radenz et al. (2018) and Williams et al. (2018). $S(v)$ denotes the spectral reflectivity in the co channel with the instrument weighting function $I(\boldsymbol{r}_0, \boldsymbol{r})$ (Doviak and Zrnic, 1993), the center of the range gate $\boldsymbol{r}_0$, the observation volume $V$, the wavelength of the radar $\lambda$ and the dielectric factor $K$

$$S(\boldsymbol{r}_0, v) = \frac{\lambda^4}{\pi^5 |K|^2} \int_V I(\boldsymbol{r}_0, \boldsymbol{r}) \eta'(\boldsymbol{r}, v) \mathrm{d}^3 \boldsymbol{r}. \tag{A1}$$

The spectrum is expressed in terms of equivalent reflectivity factor, relating the volume reflectivity $\eta'$ to the reflectivity factor $Z_e$ assuming Rayleigh scattering by water droplets. For brevity 'factor' behind reflectivity and the subscript $e$ are omitted. The cloud radar samples the Doppler spectrum at discrete velocity bins determined by the number of points in the fast Fourier transformation. Hence, $S(v)$ is represented as $S(i)$, where $v(i)$ maps the bin index to the velocity. The peak boundaries are $v_{\mathrm{left}} = v(l)$, $v_{\mathrm{right}} = v(r)$. $\overline{v}$ is the mean velocity, $\sigma$ the spectral width and $\gamma$ the skewness. For higher-order moments, tails of signal on side of the peak might cause a bias, when the other side is bound by an internal minimum (Fig. A1). To prevent this, only spectral reflectivity values $S(i)$ above the threshold that separates the (sub-)peak from its neighbor are included for calculating moments other than $Z$.

$$Z = 10 \log_{10} \sum_{i=l}^{r} S(i) \tag{A2}$$

$$\overline{v} = \frac{\sum_{i=l}^{r} S(i) v(i)}{\sum_{i=l}^{r} S(i)} \tag{A3}$$

$$\sigma^2 = \frac{\sum_{i=l}^{r} S(i) [v(i) - \overline{v}]^2}{\sum_{i=l}^{r} S(i)} \tag{A4}$$

$$\gamma = \frac{\sum_{i=l}^{r} S(i) [v(i) - \overline{v}]^3}{\sigma^3 \sum_{i=l}^{r} S(i)} \tag{A5}$$

The LDR is calculated by using the spectral reflectivity in the cross channel $S_{cx}(i)$:

$$\text{LDR} = 10 \log_{10} \frac{\sum\limits_{i=l}^{r} S_{cx}(i)}{\sum\limits_{i=l}^{r} S(i)} \tag{A6}$$

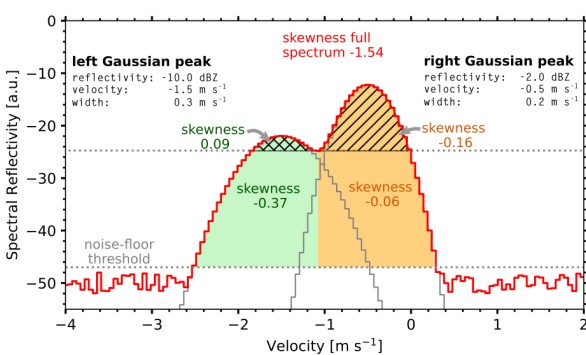

**Figure A1.** Example for different skewness values, if the spectrum is cut at the local minimum or not.

*Author contributions.* MR developed the algorithm and drafted the manuscript. JB supported the implementation. JB and PS supervised the work together. PS and HG preprocessed the Doppler spectra of MIRA-35. HG, RE and MR, operated MIRA-35 on board Polarstern. All authors jointly contributed to the manuscript and the scientific discussion.

*Competing interests.* The authors declare that they have no conflict of interest.

*Acknowledgements.* The research leading to these results has received funding from the European Union's Horizon 2020 research and innovation programme under grant agreement no. 654109 (ACTRIS), the European Union Seventh Framework Programme (FP7/2007–2013) under grant agreement no. 603445 (BACCHUS). We also acknowledge the funding by the Deutsche Forschungsgemeinschaft (DFG, German Research Foundation) – Project Number 268020496 – TRR 172, within the Transregional Collaborative Research Center „ArctiC Amplification: Climate Relevant Atmospheric and SurfaCe Processes, and Feedback Mechanisms (AC)[3]". We thank the Alfred Wegener Institute and R/V Polarstern crew and captain for their support (AWI_PS106_00) as well as the ARM AMF2 and SMEAR II teams for operating the instruments during BAECC. We gratefully acknowledge the constructive comments by Max Maahn and an anonymous referee.

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
