# Peer review of "peakTree: A framework for structure-preserving radar Doppler spectra analysis"

_Atmospheric Measurement Techniques, 2019_

## Referee Comment (RC1) · Anonymous Referee #1 · 29 Apr 2019

The authors present an algorithm for identification and detection of multiple peaks in radar Doppler spectra. This peak identification is an important step in Doppler spectral analysis in order to separate and trace different particle populations. The authors demonstrate their new algorithm with a case study of an Arctic mixed-phase cloud.

In general, I think this new algorithm is an important novel tool which will help to analyze Doppler spectra. I especially like that the authors tried to develop an algorithm which uses as few as possible a priori assumptions about the number of possible sub- or secondary peaks. In that way, it is a very flexible tool which allows the user to apply the filtering and analysis needed for her/his specific application. I also appreciate that the authors make their code publicly available which is unfortunately not the case for many previously published algorithms. I like the general structure of the manuscript

and I think the content fits nicely into AMT. However, the text, some descriptions, and the case study analysis part need some major revisions (see comments below).

General comments:

A much more careful proof-reading by all authors is needed regarding the English, punctuation, typos, and sentence structure. I will list a few examples in the specific comments but not all. This should be one of the main duties of the co-authors rather than the reviewers.

Algorithm description (Section 3): I recognize that the authors put a lot of effort in illustrating and explaining their new algorithm. However, I have to admit that I still got confused in some parts and would like to suggest a few improvements: In your example spectrum (Fig. 1) you show a spectrum with several sub-peaks but without an additional noise separated peak. I think such a more general example would be much better to illustrate the method. This would also better connect to your mixed-phase cases where one often finds the narrow, noise separated liquid peak next to the broader ice/snow peak with sometimes additional sub-peaks for example caused by riming. In such a diagram, I would also like to see all terms which are used in the text to be included. I was for example very much confused by all the node termination: root node, parent node, child node, leaf node, etc. Please make this easier for the reader to follow or to quickly figure out what is what.

I have my biggest problems with the second part of the case study (Section 4, P8, L5 and following). The description itself is very lengthy, descriptive, and contains a lot of speculations but very few clear conclusions. The fall streak analysis is also done very poorly by manually following streaks of maximum radar reflectivity. In that way, you are always tracking the largest particles which dominate the reflectivity signal. There are many datasets (Cloudnet sites, ARM database) where you can do such an analysis in a proper way and at the same time take full use of your new mode identification: Simply take the horizontal wind profile and the mean Doppler velocity of your nodes and you

can reconstruct the fall streaks of your individual nodes. The current fall streak analysis you present appears to me not very convincing. Also the application of the Hogan 2006 Z-T-IWC retrieval to the different nodes is not very sound. As you mention, Hogan et al., 2006 derived the Z-T-IWC for a large set of aircraft measured PSDs. I find it very questionable to apply such a relation to your different nodes, which you identified in order to separate (!) different particle populations and their different properties. You could have used (or even self-derived) Z-IWC retrievals for more column or needle shaped particles (for your mode with larger LDR at lower levels) and a Z-IWC retrieval for aggregates or plates for the first mode. In that way you would have demonstrated some convincing added value of your peak separating approach. I suggest to either shorten/remove some of these parts or extend it (better datasets, other cases, more appropriate Z-IWC relations).

I am also missing some discussion in your manuscript about how to best decompose Doppler spectra. Several studies in your reference list used for example Gaussian fitting or fuzzy logic while in your approach you basically cut the spectrum at the minima. I understand that your focus in this work is in the peak identification logic but I would welcome some discussion on this topic as well since it appears to me to be closely connected.

Specific comments:

Abstract, L. 2: "Cloud radar observations contain information on multiple particle species, when there are distinct peaks in the Doppler spectrum". This is not always true. Turbulence can cause multi-modal spectra even though only one population of particles is present.

Abstract, L. 3: "Complex multi-peaked situations are not captured by established algorithms". Not clear to me what you mean here. What means "complex"? What is "not captured"? What should be captured and for what? Be more specific.

P2-3, Dataset description: It appears to me that the dataset is not really ideal for

demonstrating the algorithm for Doppler spectra analysis. 10s averaging will remove a large number of interesting microphysical features and also the horizontal wind influence due to pointing uncertainties can cause many artefacts. I understand that you probably want to use data of recent campaigns to acknowledge these projects and their funding but from a scientific point of view it appears to me that there are several datasets (e.g., ARM datasets from the Arctic) which provide much better quality for such a demonstration.

Figure 1: Why does the spectrum have these "tails" to the sides (lowest/fastes velocities). It looks like a broadening effect due to the long temporal averaging and/or swinging of the beam with the ship motion.

P3, L12: "with signal above the noise level": Please provide the exact threshold when you consider the signal to be above the noise.

P3, L20: I can't find v_left/right/add in Fig. 1. Are they not relevant for the algorithm? As I mentioned in my general comments, it would be good to show an example which contains a noise-separated peak. Here, you only describe it but in such an example you could easily explain all terms used.

Figure 1a: I suggest to remove the "units" of the spectral reflectivity (dBZ) and rather use arbitrary units [a. u.] or [dB]. If you would plot the spectrum in linear units, you could write (mm^6/m^3)/(m/s). In that way, the integral over the full spectrum would result in the usual linear units of Ze. However, the integral over a log spectrum will neither result in mm^6/m^3 nor dBZ. The radar experienced readers will certainly understand what you mean but it's simply not correct in a strict scientific sense.

Figure 1b: It is not clear to me how I can read the skewness from the triangle, please explain. The caption is also missing the description of what is meant with "spec Z cx" or the line "decoupling".

P3, L32-33: "Only the part of the Doppler spectrum above the threshold defined by the

spectral reflectivity minimum that separated the peaks are used". This is a problematic aspect of your approach which I think should be discussed much more and maybe even changed. Let's consider only Ze: For Node 0 you integrate the full spectrum starting at the noise level. Already for Node 1, you integrate only starting from your first threshold (-34 dB). I don't understand why you are not integrating again from the noise level? I would expect that when I sum up all the identified sub-peaks (Node4+3+2; I exclude Node 1 since it is basically 3+4), the resulting Ze should be identical to Node 0. But if I understand correctly, this is not the case for your algorithm, or? From a microphysical point of view, I guess one would like to have moment estimates of the full sub-peak and not only the "peak head" which sticks out of the remaining spectrum.

P4, L3-4: Where do I find Node 5 and 6 in Fig. 1?

P6, Title Section 4: Replace "ice crystal habits" with "ice crystal populations". The spectra indicate that you have two populations of particles with different fall velocities. This could be related to different habits but you could also have two populations with different fall speed and similar habits (e.g. due to onset of riming).

P6, L9: "humidity profile": Actually you only show profiles of air temperature and dew point. The humidity information is contained in them but why not plotting relative humidity directly?

P7, L2-3: "previous studies used the simple criterion of low reflectivity and vertical velocity close to $0ms^{-1}$ to identify regions of a cloud, where the presence of liquid is likely" I think this description is not very precise: In fact, the peak is thought to be due to liquid if it is a very narrow peak since the PSD of super-cooled droplets can be assumed to be rather narrow. In the way you describe it, any peak with low Ze and v close to 0 m/s might be interpreted as liquid. How reliable are those thresholds (especially the Ze threshold)? Are your values different from the studies cited? Are the thresholds used within those studies all the same or different?

Figure 3f: Why is there no color for N=2? A second node would be the most likely

scenario for a liquid water and an ice peak, or?

P8, L1+6: Be consistent whether you use the minus sign when indicating the Doppler velocity or not.

P8, L1: The low LDR indicates plate-like particles, right? But then they are oblate and not prolate (like columns). At P12, L11 you denote them as oblate...

P9, L24: "indicating no change in particle habit": Well, if the particle habit changes for example from plates to dendrites, I would also not expect a big change in LDR. I think the conclusion that habit does not change only because LDR is rather constant is not true in general.

P9: In addition to my principal problems with your fall streak analysis (see general comments): Why don't you show range spectrograms for your different fall streaks?

P12, L4: Another important advantage of your method to microARSCL is that you provide the code for the community. For further development of Doppler spectra analysis, this is absolutely key!

P13, L7: Why are v_left/right relevant for the moment estimation. They don't appear in any formulas. How are they actually determined? Maybe a certain threshold for the spectrum above the noise level?

Style and Typos:

Abstract, L. 3: Add comma after "In this study" and before "that". These are very typical punctuation mistakes which I found very often throughout the manuscript. I will not list them all but ask all authors to do a more careful reading.

P1, L15: Better: "Cloud radars are frequently used..." P1, L17: Add comma after "In general" P2, L.4: Add comma "formed ice, and" P2, L7: "In a further step," P2, L7-9: Confusing and very complicated sentence. Please re-structure and/or split in two. P2, L10: Prior approached should be approaches P2, L12: Remove comma

after "available" P2, L14: "In this study," P2, L16: "algorithm IS easily applicable" P2, L24: More a question to the editors but are citations of manuscripts in preparation appropriate? P3, L20, L31: "In a first step," "In the next step," P3, L23: Add comma after v_add P3, L22: Better "All minima found" P7, L1: Remove comma before "that" P7, L2 and P9, L3: "can not" vs "cannot" use consistently P7, L7: "during the whole case study" better "during the entire event" P7, L7/L8: "The top/second one" is a bit slang-like, better "The uppermost layer" P7, L7: "single moments of the full spectrum": I think the "single" is redundant here P7, L9-10: "Together with the lidar backscatter indicating a liquid cloud base at 750m between" awkward sentence, please rephrase. P9, L4-7: Very long and complicated sentence. Split in two and rephrase. Also, the sentence is very speculative. P9, L1: add comma before "which" P12, L6: application OF this new P12, L6: In a second step, P12, L13: Within this liquid layer,

---

## Referee Comment (RC2) · Anonymous Referee #2 · 10 May 2019

The authors present a novel method to describe multiple peaks in the radar Doppler spectrum. Personally, I thought about this for a couple of years and I am very excited to see that there is finally a good idea and progress. In particular, I like that there is no need for a somewhat arbitrary distinction between noise separated and sub-peaks anymore. I recommend the manuscript to be published, but I think the manuscript can be strongly improved by addressing the following points:

Major comments

Impact of 10s averaging: 10s is quite a lot for spectral cloud radar applications. Most cloud radar data set I'm aware of (e.g. ARM) use temporal resolutions in the order of 1-3 s. What is the impact of this on the method? I would expect that the spectrum is bumpier when averaging less and that maybe a different sub-peak threshold needs to

be used? At the same time, sub-peaks might get smoothed out during averaging. And do I assume correctly that the authors hope that the ship motion cancels out within 10 s?

Using Doppler spectrum above the separation threshold for moment estimation: I think I understand why the authors decided to estimate the moments this way. However, I'm afraid that this method will also lead to biases, in particular for the higher moments. The authors could do a quick sensitivity study and quantify the change of the moments by using a normal monomodal peak and cutting of the tails at different spectral reflectivities.

Grouping: The authors should discuss why they chose the threshold used for the grouping (50s, 150 m, d<0.9, 0.4 and 0.9 normalization factors) and the impact of changing these thresholds on the results.

Application of grouping: I would strongly recommend adding a few sentences on how the data set was grouped exactly. I do not understand a couple of processing steps: Did the authors manually select anchor nodes for both nodes separately? What criteria were used? Were the criteria for liquid nodes used in Fig 4 used as a starting point for one node? What about other nodes then the two shown ones? Can they be grouped, too?

Language: The paper needs a lot of work to improve the English. German grammatical structure shows through in numerous places. Word selection and punctuation can also be improved upon. Given that publishing includes language editing for Copernicus, I do not list language-related issues.

Minor comments

P1L19: I would recommend adding a short discussion about the difference between peaks that are separated by noise and peaks which are not (eg see fig 13 of Williams et al 2018)

[Figure]

P2L2 "which likely causes significant errors": Are the authors sure? I would argue that for most empirical retrievals the climatology of multi-peak situations is (unknowingly) included into the retrieval so that there are no biases. Also applies to P9L30.

P2L23: I don't think the authors can cite papers in preparation

P2L29: vertical-stare -> vertically pointing

P2L30: This part could be shortened using a table with the radar specifications.

P3L12: LDR -> LDR spectrum?

P3L28f "the prominence of one of its subpeaks is less than 1 dB" and "height of the peak above": I assume the authors talk about the maximum of the subpeak?

P3L32 "Doppler spectrum above the threshold": In the appendix, the authors mention this does not apply to Ze?

P4 Fig1a: According to P3L19f, node 0 is the full radar Doppler spectrum

P4 Fig1b: An explanation of how skewness is actually displayed in the figure is missing.

P4L1: node 0 -> node 1?

P5 Table1: Z is not defined yet. Also, I strongly recommend to use Ze (equivalent radar reflectivity factor) instead of Z (radar reflectivity factor) because Z is typically defined with $10*\log_{10}(\text{SUM}(N*D^6))$ which applies only to Rayleigh scattering of liquid drops (see eg. 'Radar for Meteorologists' by R. Rinehart.

P5L1: I would recommend indicating that 'indices' refers to the nodes not the bin in the Doppler spectrum.

P6 Fig2: Are Ze and v normalized in this plot?

P6L14: 'giving hints' please specify

P7 Fig3: I would recommend adding the fallstreaks also to this figure because it is

interesting that the lower end of the first one can be only seen after applying the grouping.

P7L3 "to identify regions of a cloud, where the presence of liquid is likely": I would say these thresholds are rather to identify regions where drops are the dominating particle type. Liquid is likely also present in other cloud regions.

P7L9 "periods of liquid" add 'likely' or 'possible'

P7L9f: The liquid extends from 750 to 1000 m?

P7L13 "The faster-falling particle population" and title Fig 5: was velocity or LDR used or grouping? Also, I would recommend naming the nodes consistently.

P7L15 "generated ice": the authors should mention before that they assume the second peak to be ice

P8 Fig 4: Because it is described in the section before, I would recommend to clearly indicate that the grouping is not used in this figure.

P8 Fig 4: I would recommend indicating the ceilometer cloud base by e.g. a black in this and other plots

P8 Fig 4: How does LDR look? It should have a sufficient SNR at least for the lower layer.

P9L3 "We cannot fully rule out that ice multiplication was triggered...": the authors should think about removing the following discussion because it is speculation and not of importance for this study.

P9L14ff: It is a little challenging to follow which population the authors discuss.

P10 Fig 5: How many anchors were manually selected here?

P10 Fig 6: P10 Fig 6: Given that IWC scales with Ze, I don't see a benefit of this figure.

P12L4: To my knowledge, microARSCL uses actually sub. I would recommend to focus

stronger on the greater flexibility by overcoming the separation in noise separated and sub peaks.

Appendix A: I would recommend adding a definition for spectral reflectivity.

P13L19: MIRA or Mira-35?

---

## Author Comment (AC1) · 16 Jul 2019

**General Remarks**

We thank both referees for carefully reading the manuscript and providing clear and constructive comments for improving it. We highly appreciate their willingness to review a revised version of the manuscript. To our impression, most confusion and misunderstanding was caused by insufficient description, what should be addressed by our algorithm: the peak structuring. The simplifications made at prior (peak identification) and later steps (peak interpretation) were made to not distract from the major focus. This was not clearly communicated. We tried to be more comprehensible and precise in the formulation and included more material. The fundamental changes are briefly described here, before each point is addressed individually below. The referee comments are marked in grey and the response in black with indentation.

We revised the introduction and discussion section, to make the scope of this study clearer. More emphasis is put on the fact, that the novelty of our approach is the peak structuring part. The peak identification/finding and the interpretation step are kept rather straightforward on purpose. A second example Doppler spectrum was added to the algorithm description.

Both referees raised doubts about the quality of the dataset. We are confident, that the quality of the dataset is sufficient, especially when taking into account the conditions under which it was sampled (shipborne in the Arctic). However, we comprehend the problems that might arise from introducing a new method on an imperfect dataset. Hence, we decided to add a second application case study based on data from the ARM campaign BAECC at Hyytiälä, Finland.

We want to emphasize, that the algorithm itself is independent of the averaging time. We choose the 10 seconds average for the PS106 dataset for two reasons: Firstly, to smooth out the uncertainties arising from pointing uncertainties due to ship motions (especially after the breakdown of the stabilization platform). Fig 1. below illustrates this issue. Secondly, we wanted to be sure, that a detected local minimum in the spectral reflectivity is not caused by noise, i.e. reduce the number of false positives. One approach to reduce the uncertainty in the Doppler spectrum estimate is to sample more realizations and average [Zrnic 1975 JAM]. The standard deviation per spectral bin will reduce with $1/\sqrt{N}$. For our settings of 5kHz pulse repetition frequency, 256 FFT length and no coherent averages, the expected standard deviation per bin is 0.3 dB, which is sufficiently below the 1dB prominence threshold. Also work by other authors (as well dealing with KAZR data) had to do additional smoothing to get useful information from the Doppler spectrum. E.g. Luke and Kollias 2013 [JTECH] used a 20 second running window and Kalesse et al. 2019 [AMTD] used 18 seconds averages.

In the frame the averaging might hide few spurious features, but from our experience, the features we are interested in are persistent over times longer than the averaging time under such smooth and stratiform conditions as observed here.

[Figure]

**Fig 1 MIRA-35 vertical velocity in different temporal resolutions as well as pitch and roll angle measured by the ships navigation system**

**Specific Reply to Referee #1**

A much more careful proof-reading by all authors is needed regarding the English, punctuation, typos, and sentence structure. I will list a few examples in the specific comments but not all. This should be one of the main duties of the co-authors rather than the reviewers.

> Thanks for this comment. We will consult a native English speaker before submission of the final version of the manuscript.

Algorithm description (Section 3): I recognize that the authors put a lot of effort in illustrating and explaining their new algorithm. However, I have to admit that I still got confused in some parts and would like to suggest a few improvements: In your example spectrum (Fig. 1) you show a spectrum with several sub-peaks but without an additional noise separated peak. I think such a more general example would be much better to illustrate the method. This would also better connect to your mixed-phase cases where one often finds the narrow, noise separated liquid peak next to the broader ice/snow peak with sometimes additional sub-peaks for example caused by riming. In such a diagram, I would also like to see all terms which are used in the text to be included. I was for example very much confused by all the node termination: root node, parent node, child node, leaf node, etc. Please make this easier for the reader to follow or to quickly figure out what is what.

> Thanks for the suggestion. We have added a sketch (new Fig. 1) to illustrate the indices and naming conventions.
> The figure illustrating the tree generation (new Figure 2) now contains two spectra. One including a noise floor separated peak and 9 nodes and the second one where spectral LDR is available.

I have my biggest problems with the second part of the case study (Section 4, P8, L5 and following). The description itself is very lengthy, descriptive, and contains a lot of speculations but very few clear conclusions. The fall streak analysis is also done very poorly by manually following streaks of maximum radar reflectivity. In that way, you are always tracking the largest particles which dominate the

reflectivity signal. There are many datasets (Cloudnet sites, ARM database) where you can do such an analysis in a proper way and at the same time take full use of your new mode identification: Simply take the horizontal wind profile and the mean Doppler velocity of your nodes and you can reconstruct the fall streaks of your individual nodes. The current fall streak analysis you present appears to me not very convincing.

> We now emphasize more clearly in the text that we used a simplified fallstreak tracking procedure. References to more sophisticated methods are given in the text (now: P12, L3-15). However, we also need to note here that the fallstreak tracking based on the horizontal wind profiles works only in case of absence of directional wind shear. In the presented case, the wind direction in the height range covered by the tracked fallstreaks varied between from 230° to 300° and back to 150° (Fig. 2 below).

[Figure]

Fig 2: Hodograph from the sounding shown in the manuscript (29 June 2017 10:50 UTC).

Also the application of the Hogan 2006 Z-T-IWC retrieval to the different nodes is not very sound. As you mention, Hogan et al., 2006 derived the Z-T-IWC for a large set of aircraft measured PSDs. I find it very questionable to apply such a relation to your different nodes, which you identified in order to separate (!) different particle populations and their different properties. You could have used (or even self-derived) Z-IWC retrievals for more column or needle shaped particles (for your mode with larger LDR at lower levels) and a Z-IWC retrieval for aggregates or plates for the first mode. In that way you would have demonstrated some convincing added value of your peak separating approach. I suggest to either shorten/remove some of these parts or extend it (better datasets, other cases, more appropriate Z-IWC relations).

> We agree that the Hogan 2006 retrieval is not optimal here, because it is designed for a single particle species (aggregates) and the underlying dataset was derived in deep clouds. In Cloudnet, for which the Hogan 2006 retrieval was originally designed, the retrieval is usually applied to all clouds, independent of their nature. In our case, we see the precipitation from two stratiform clouds overlapping. Hence, the principle application is nearly the same as usual, taking into account the error intervals for the temperature range given in Hogan et al. (2006). In the context of this paper, the IWC values are presented to give an impression on how the mass relation is between the single peaks. This is actually only possible if peaks were successfully separated first. Nevertheless, we decided to modify Fig 9 (former Fig. 6) in such a way that it now presents the total IWC (node 0) and the ratio of the IWC of the selected node to the total IWC. This should give a better impression of the applicability of peakTree for the

investigation of the microphysics of different nodes. For future work, we would like to use other retrievals based on the actual shape of the particles.

I am also missing some discussion in your manuscript about how to best decompose Doppler spectra. Several studies in your reference list used for example Gaussian fitting or fuzzy logic while in your approach you basically cut the spectrum at the minima. I understand that your focus in this work is in the peak identification logic but I would welcome some discussion on this topic as well since it appears to me to be closely connected.

We would consider the focus of the algorithm to be on peak structuring. From our point of view, the peak decomposition step should be called peak identification.

However, the performance of the fuzzy logic, continuous wavelet transform or fitting techniques are more powerful, if the peaks that are noise-floor or local-minimum separated are segregated beforehand. Then these techniques could focus on identifying hidden sub-peaks in a mono-modal appearing peak.

We rewrote the respective section in the introduction to make this issue clearer.

Abstract, L. 2: "Cloud radar observations contain information on multiple particle species, when there are distinct peaks in the Doppler spectrum". This is not always true. Turbulence can cause multi-modal spectra even though only one population of particles is present.

Thank you for pointing this out. We weakened the statement by including 'frequently'.

Abstract, L. 3: "Complex multi-peaked situations are not captured by established algorithms". Not clear to me what you mean here. What means "complex"? What is "not captured"? What should be captured and for what? Be more specific.

'Complex' is omitted, as it is implicitly included into the 'multi-peak' statement. 'not captured' was replaced by 'not taken into account'.

P2-3, Dataset description: It appears to me that the dataset is not really ideal for demonstrating the algorithm for Doppler spectra analysis. 10s averaging will remove a large number of interesting microphysical features and also the horizontal wind influence due to pointing uncertainties can cause many artefacts. I understand that you probably want to use data of recent campaigns to acknowledge these projects and their funding but from a scientific point of view it appears to me that there are several datasets (e.g., ARM datasets from the Arctic) which provide much better quality for such a demonstration.

As both referees raised this concern, we discussed it above in the General remarks section.

Figure 1: Why does the spectrum have these "tails" to the sides (lowest/fastest velocities). It looks like a broadening effect due to the long temporal averaging and/or swinging of the beam with the ship motion.

We are puzzled by these tails as well. So far we could exclude effects from averaging, ship movement and FFT windowing. Currently we are in contact with the manufacturer, who presumes a rather low level technical issue.

P3, L12: "with signal above the noise level": Please provide the exact threshold when you consider the signal to be above the noise.

We cannot provide one threshold, as noise also depends also on range. Nevertheless, we refined the mentioned sentence: 'Hence, when calculating the LDR (Eq. A6) only bins where the signals in the cross channel is a factor of 3 above the noise level are taken into account.'

P3, L20: I can't find v_left/right/add in Fig. 1. Are they not relevant for the algorithm? As I mentioned in my general comments, it would be good to show an example, which contains a noise-separated peak. Here, you only describe it but in such an example, you could easily explain all terms used.

> We added a noise-floor separated spectrum and the descriptions into the figure, which is now Fig. 2.

Figure 1a: I suggest to remove the "units" of the spectral reflectivity (dBZ) and rather use arbitrary units [a. u.] or [dB]. If you would plot the spectrum in linear units, you could write (mm^6/m^3)/(m/s). In that way, the integral over the full spectrum would result in the usual linear units of Ze. However, the integral over a log spectrum will neither result in mm^6/m^3 nor dBZ. The radar experienced readers will certainly understand what you mean but it's simply not correct in a strict scientific sense.

> Thanks for pointing to this issue. We settled to [a.u.] (now in new Figure 2)

Figure 1b: It is not clear to me how I can read the skewness from the triangle, please explain. The caption is also missing the description of what is meant with "spec Z cx" or the line "decoupling".

> The triangle is only a qualitative indication for skewness, we included a clarification as well as a description of the mentioned terms to the caption (now Figure 2).

P3, L32-33: "Only the part of the Doppler spectrum above the threshold defined by the spectral reflectivity minimum that separated the peaks are used". This is a problematic aspect of your approach which I think should be discussed much more and maybe even changed. Let's consider only Ze: For Node 0 you integrate the full spectrum starting at the noise level. Already for Node 1, you integrate only starting from your first threshold (-34 dB). I don't understand why you are not integrating again from the noise level? I would expect that when I sum up all the identified sub-peaks (Node4+3+2; I exclude Node 1 since it is basically 3+4), the resulting Ze should be identical to Node 0. But if I understand correctly, this is not the case for your algorithm, or? From a microphysical point of view, I guess one would like to have moment estimates of the full sub-peak and not only the "peak head" which sticks out of the remaining spectrum.

> Thanks for pointing out this ambiguous formulation. The comment addresses, what we tried to explain in the Appendix with 'To prevent this, only spectral reflectivity values S(i) above the threshold that separates the subpeak from its neighbor are included for calculating moment other than Z.'
> We added Fig A1 to illustrate, why it would introduce a bias in higher order moments, when integrating over the whole peak (-0.37 vs 0.09 for the left peak). However, for the reflectivity, the concerns are valid. The revised explanation reads: 'Reflectivity Z is calculated by integrating the spectral reflectivity of the whole peak (i.e. from the noise floor up). For all higher moments, signal below the threshold, that separated the (sub-)peak is neglected to avoid biases (see also Fig. A1)'

P4, L3-4: Where do I find Node 5 and 6 in Fig. 1?

> Nodes 5 and 6 would exist if node 2 (the rightmost sub-peak) would have two sub-peaks. We have added a sketch (new Fig. 1) with possible indices in the beginning of the algorithm description.

P6, Title Section 4: Replace "ice crystal habits" with "ice crystal populations". The spectra indicate that you have two populations of particles with different fall velocities. This could be related to different habits but you could also have two populations with different fall speed and similar habits (e.g. due to onset of riming).

> Done as suggested (P9,L19).

P6, L9: "humidity profile": Actually you only show profiles of air temperature and dew point. The humidity information is contained in them but why not plotting relative humidity directly?

> We replaced the term 'humidity profile' by 'spread between temperature and dewpoint' (P9,L25). However, we decided to stick to dewpoint and temperature as plotted variables (now Fig. 6). Firstly, dewpoint is a more direct measure for moisture content, as relative humidity depends strongly on air temperature. Secondly, relative humidity would have required a new subfigure or at least new x-axis, which would make the plot more difficult to read.

P7, L2-3: "previous studies used the simple criterion of low reflectivity and vertical velocity close to 0ms^-1 to identify regions of a cloud, where the presence of liquid is likely" I think this description is not very precise: In fact, the peak is thought to be due to liquid if it is a very narrow peak since the PSD of super-cooled droplets can be assumed to be rather narrow. In the way you describe it, any peak with low Ze and v close to 0 m/s might be interpreted as liquid. How reliable are those thresholds (especially the Ze threshold)? Are your values different from the studies cited? Are the thresholds used within those studies all the same or different?

> We use the same thresholds as Oue 2018. Kalesse 2016 also reported similar values. Yu 2014 and Frisch 1995 used slightly higher reflectivity thresholds. Our choice of -20dBZ is rather conservative compared to these studies. Judging, how reliable the threshold are is beyond the scale of this study. However, the good agreement with the ceilometer cloud base, especially in the BAECC case makes us quite confident.

Figure 3f (new Fig. 6f): Why is there no color for N=2? A second node would be the most likely scenario for a liquid water and an ice peak, or?

> A full binary tree with two nodes is not possible. Either, the tree contains only the root node (mono-modal spectrum) or left and right sub-peak, hence 3 nodes in total. I.e. the situation of ice and liquid water peak will be N=3.

P8, L1+6: Be consistent whether you use the minus sign when indicating the Doppler velocity or not.

> Done as suggested.

P8, L1: The low LDR indicates plate-like particles, right? But then they are oblate and not prolate (like columns). At P12, L11 you denote them as oblate.

> 'oblate' is the correct term in this context. Thanks for mentioning.

P9, L24: "indicating no change in particle habit": Well, if the particle habit changes for example from plates to dendrites, I would also not expect a big change in LDR. I think the conclusion that habit does not change only because LDR is rather constant is not true in general.

> The argument of the referee is perfectly right for the general case. For this specific case LDR values of -14dB can only be caused by strongly prolate particles (like columnar ice needles). Any change towards more oblate particles due to aggregation or riming would decrease LDR. We modified the statement, to make this special case clearer (P13,L5): "During this growth, LDR remains at the high value -14 dB, indicating no change of the prolate particle shape."

P9: In addition to my principal problems with your fall streak analysis (see general comments): Why don't you show range spectrograms for your different fall streaks?

> We decided to not show the spectrograms because this would have increased the amount of Figures without providing additional information. To our notion, the general nature of a

spectrogram is already depicted in the moment plots of the two main nodes in Fig 8 (formerly Fig. 6).

P12, L4: Another important advantage of your method to microARSCL is that you provide the code for the community. For further development of Doppler spectra analysis, this is absolutely key!

Thanks for emphasizing this.

P13, L7: Why are v_left/right relevant for the moment estimation. They don't appear in any formulas. How are they actually determined? Maybe a certain threshold for the spectrum above the noise level?

v_left and v_right appear in the formuals implicitly via the summation indices i=l to r. The relation between v_left and l as well v_right and r are described in the paragraph above.

Abstract, L. 3: Add comma after "In this study" and before "that". These are very typical punctuation mistakes, which I found very often throughout the manuscript. I will not list hem all but ask all authors to do a more careful reading.

Thanks for the hint. We corrected the error.

P1, L15: Better: "Cloud radars are frequently used.." P1, L17: Add comma after "In general" P2, L.4: Add comma "formed ice, and" P2, L7: "In a further step," P2, L7-9: Confusing and very complicated sentence. Please re-structure and/or split in two. P2, L10: Prior approached should be approaches P2, L12: Remove comma after "available" P2, L14: "In this study," P2, L16: "algorithm IS easily applicable" P2, L24: More a question to the editors but are citations of manuscripts in preparation appropriate? P3, L20, L31: "In a first step," "In the next step," P3, L23: Add comma after v_add P3, L22: Better "All minima found" P7, L1: Remove comma before "that" P7, L2 and P9, L3: "can not" vs "cannot" use consistently P7, L7: "during the whole case study" better "during the entire event" P7, L7/L8: "The top/second one" is a bit slang-like, better "The uppermost layer" P7, L7: "single moments of the full spectrum": I think the "single" is redundant here P7, L9-10: "Together with the lidar backscatter indicating a liquid cloud base at 750m between" awkward sentence, please rephrase. P9, L4-7: Very long and complicated sentence. Split in two and rephrase. Also, the sentence is very speculative. P9, L1: add comma before "which" P12, L6: application OF this new P12, L6: In a second step, P12, L13: Within this liquid layer,

Corrected all mentioned errors. Thanks a lot for the effort!

**Specific Reply to Referee #2**

Impact of 10s averaging: 10s is quite a lot for spectral cloud radar applications. Most cloud radar data set I'm aware of (e.g. ARM) use temporal resolutions in the order of 1-3 s. What is the impact of this on the method? I would expect that the spectrum is bumpier when averaging less and that maybe a different sub-peak threshold needs to be used?

The generation of the tree itself is independent of the averaging time. However, a noisier 'bumpier' spectrum would cause numerous narrow nodes, which would increase the effort necessary for post processing, especially peak interpretation.

At the same time, sub-peaks might get smoothed out during averaging. And do I assume correctly that the authors hope that the ship motion cancels out within 10s?

As both referees raised this concern, we discussed it above in the General remarks section.

Using Doppler spectrum above the separation threshold for moment estimation: I think I understand why the authors decided to estimate the moments this way. However, I'm afraid that this method will

also lead to biases, in particular for the higher moments. The authors could do a quick sensitivity study and quantify the change of the moments by using a normal monomodal peak and cutting of the tails at different spectral reflectivities.

> Currently, we separate the peaks above their floor to get rid of basic systematic errors introduced into the calculation of the skewness when considering a peak which is cut off at one or both sides. To our knowledge no better approach to deal with such situations is available yet. The skewness measured above the threshold can only be compared to the original skewness in a very limited way and must be treated with caution. It will probably have the same sign, but the magnitude will be different, because the numeric value of skewness reacts very sensitively to changes in the outer parts of a peak. We mention this now in the paper in Section 3.1 and in Appendix 1.

Grouping: The authors should discuss why they chose the threshold used for the grouping (50s, 150 m, d<0.9, 0.4 and 0.9 normalization factors) and the impact of changing these thresholds on the results. Application of grouping: I would strongly recommend adding a few sentences on how the data set was grouped exactly. I do not understand a couple of processing steps: Did the authors manually select anchor nodes for both nodes separately? What criteria were used?

> We admit, that the choice of the thresholds seems arbitrary on the first look. They are the result of manual and iterative interpretation. The quality criterion for the grouping is the consistency and smoothness of the moments in time and range for each particle population. A new paragraph in the 'Discussions and conclusions' section discusses the impact of each threshold.  We want to emphasize again, that this grouping approach is no prerequisite of the peak structuring algorithm, but an example how this structure can be used to interpret peaks.

Were the criteria for liquid nodes used in Fig 4 used as a starting point for one node?

> The criteria for the liquid nodes were not used for this interpretation. But during the data analysis itself, the two detected liquid layers triggered a more thorough investigation of this feature.

What about other nodes then the two shown ones? Can they be grouped, too?

> In principle, they could be grouped, too. This second grouping step would only provide additional information for trees with more than 3 nodes. Also for the BAECC case grouping might be an interesting analysis, but is beyond of the scope of this study.

Language: The paper needs a lot of work to improve the English. German grammatical structure shows through in numerous places. Word selection and punctuation can also be improved upon. Given that publishing includes language editing for Copernicus, I do not list language-related issues.

P1L19: I would recommend adding a short discussion about the difference between peaks that are separated by noise and peaks which are not (eg see fig 13 of Williams et al 2018)

> From our point of view, there is no conceptual difference between noise-floor separated peaks and peaks only distinguishable by a minimum in spectral reflectivity. We added a paragraph on this issue in the discussion (P16,L1-7).

P2L2 "which likely causes significant errors": Are the authors sure? I would argue that for most empirical retrievals the climatology of multi-peak situations is (unknowingly) included into the retrieval so that there are no biases. Also applies to P9L30.

We here provide a reply that we also gave to a similar comment of Referee 1: In the context of this paper, the IWC values are presented to give an impression on how the mass relation is between the single peaks. This is actually only possible if peaks were successfully separated first. Nevertheless, we decided to modify Fig 9 (former Fig. 6) in such a way that it now presents the total IWC (node 0) and the ratio of the IWC of the selected node to the total IWC. This should give a better impression of the applicability of peakTree for the investigation of the microphysics of different nodes. For future work, we would like to use other retrievals based on the actual shape of the particles.

P2L23: I don't think the authors can cite papers in preparation

The paper will be submitted by the end of the review process. Basically the paper of Griesche et al. is a considerable extension in comparison to the information about the instruments given in Wendisch et al. (2018).

P2L29: vertical-stare -> vertically pointing

changed

P2L30: This part could be shortened using a table with the radar specifications.

A new Table 1 with the radar specifications (now for both radars) was added.

P3L12: LDR -> LDR spectrum?

Here the 'bulk' LDR is meant. The LDR spectrum is not used at all, only the co and cross channel spectrum.

P3L28f "the prominence of one of its subpeaks is less than 1 dB" and "height of the peak above": I assume the authors talk about the maximum of the subpeak?

Thanks for pointing out that sloppy formulation. We have refined the paragraph (P5,L6 and further): 'A minimum is skipped, if the prominence of either of its subpeaks is less than 1dB. Prominence is the difference between the maximum spectral reflectivity of a subpeak and the threshold that is defined as by the spectral reflectivity at local minimum (dashed grey lines in Fig 2 (a); similar to Shupe et al., 2004).'

P3L32 "Doppler spectrum above the threshold": In the appendix, the authors mention this does not apply to Ze?

Referee 1 addressed this ambiguous explanation as well. We added Fig A1 to the Appendix in order to illustrate why it would introduce a bias in higher order moments, when integrating over the whole peak (-0.37 vs 0.09 for the left peak). However, for the reflectivity, the given concerns are valid. The revised explanation reads (P5,L10f): 'Reflectivity factor Z is calculated by integrating the spectral reflectivity of the whole peak (i.e. from the noise floor up). For all higher moments, signal below the threshold, that separated the (sub-)peak is neglected to avoid biases (see also Fig. A1)'

P4 Fig1a: According to P3L19f, node 0 is the full radar Doppler spectrum

More clearly, it is the full Doppler spectrum above the Hildebrand Sekhon noise threshold. We refined the formulation in the text. We have rephrased the algorithm description, the mentioned sentence now reads (P4,L15-16): "The root node contains all signal of Doppler spectrum above the noise threshold between -v_Nyq and +v_Nyq." The Nyquist velocities v_Nyq of the involved cloud radars are given in Tab. 1.

P4 Fig1b: An explanation of how skewness is actually displayed in the figure is missing.

Thanks for pointing to that issue. We modified the figure caption (now Fig. 2) accordingly. 'Spectral width and skewness are shown by grey lines and triangles, respectively' now reads 'Spectral width is indicated quantitatively by the length of the grey lines and sign of the skewness is indicated by a triangle (pointing to left for negative skewness and vice-versa).'

P4L1: node 0 -> node 1?

No, node 0 (or the root node) is correct in this context.

P5 Table1: Z is not defined yet. Also, I strongly recommend to use Ze (equivalent radar reflectivity factor) instead of Z (radar reflectivity factor) because Z is typically defined with $10*log10(SUM(N*D^6))$ which applies only to Rayleigh scattering of liquid drops (see eg. 'Radar for Meteorologists' by R. Rinehart.

Thanks for addressing this point. The equivalent radar reflectivity was already used implicitly. We added a sentence in Appendix 1 to clarify.

P5L1: I would recommend indicating that 'indices' refers to the nodes not the bin in the Doppler spectrum.

Thanks for the suggestion, the column title now is 'Node index'.

P6 Fig2: Are Ze and v normalized in this plot?

No, the plot (now in Fig. 3) shows the actual values. Only for calculating the Eucledian distance d, the normalization described in the text is used.

P6L14: 'giving hints' please specify

We added ", such as size or shape." to the end of the sentence (P9,L31).

P7 Fig3: I would recommend adding the fallstreaks also to this figure because it is interesting that the lower end of the first one can be only seen after applying the grouping.

The fallstreaks were added to this figure (now Fig. 6).

P7L3 "to identify regions of a cloud, where the presence of liquid is likely": I would say these thresholds are rather to identify regions where drops are the dominating particle type. Liquid is likely also present in other cloud regions.

Thanks for mentioning. We moved this paragraph to a dedicated subsection (3.2.1) in 'Algorithm' section and rephrased it. However, the drops are not required to be the dominating particle type, they only need to cause their own (sub-)peak.

P7L9 "periods of liquid" add 'likely' or 'possible'

Done as suggested.

P7L9f: The liquid extends from 750 to 1000 m?

Most likely, as there is also a layer of high humidity indicated by the temperature/dewpoint profile from the sounding at these heights (see Fig. 6).

P7L13 "The faster-falling particle population" and title Fig 5: was velocity or LDR used or grouping? Also, I would recommend naming the nodes consistently.

For the manual assignment of the anchor nodes reflectivity, velocity and LDR was taken into account. The automated grouping only used velocity and reflectivity.

P7L15 "generated ice": the authors should mention before that they assume the second peak to be ice

> This sentence refers to the ice formed at cloud top, where no second peak is present. We modified the sentence to make this clearer (P11,L4f): 'Below 2.5 km height, the ice particles generated at cloud top descent with velocities of [...]'

P8 Fig 4: Because it is described in the section before, I would recommend to clearly indicate that the grouping is not used in this figure.

> Indeed, this was confusing in the original manuscript. We moved the description of the liquid node selection and the grouping into separate subsections of the Algorithm section and referred to the respective subsection in the caption of now Figure 7 (former Fig. 4).

P8 Fig 4: I would recommend indicating the ceilometer cloud base by e.g. a black in this and other plots

> Thanks for the suggestion. We added the cloud bases for both case studies (now Figures 4, 5, 6, 7 and 8).

P8 Fig 4: How does LDR look? It should have a sufficient SNR at least for the lower layer.

> The referee raises an interesting question. LDR for the liquid peak is shown in Fig. 3 below. In the topmost part of the lower layer (now shown in Fig. 7), where the most liquid water should reside, SNR is not sufficient to detect LDR. However, in the lower part of the layer the node (or subpeak) ice particles are likely contributing to this peak, making the LDR rather high. We expect that this is the case because the depolarizing prolate ice particles only affect the skewness of the respective node because freshly formed columns have vertical motions which are similar to the ones of the liquid droplets. We decided to not discuss this further in the manuscript.

[Figure]

**Fig 3 Linear depolarization ratio of the node detected as liquid containing (as Fig 7 in the new manuscript or Fig 4 in the old manuscript). Regions with insufficient signal for LDR estimation are marked in grey.**

P9L3 "We cannot fully rule out that ice multiplication was triggered...": the authors should think about removing the following discussion because it is speculation and not of importance for this study.

We are confident, that the peakTree approach will be of considerable benefit for future studies of the microphysical structure of clouds. In order to emphasize this notion, we decided to add some discussion about possible microphysical processes to the text.

P9L14ff: It is a little challenging to follow which population the authors discuss.

Thanks for pointing to that hard to follow paragraph. We have added links to the discussed particle population (P12 L9, P12 L12, P13 L2, P13 L4).

P10 Fig 5: How many anchors were manually selected here?

As stated in the text, an anchor node every 50s and 150m. For Fig. 8 (former Fig. 5) it's then 720 nodes.

P10 Fig 6: P10 Fig 6: Given that IWC scales with Ze, I don't see a benefit of this figure.

We would like to point to the answer to Referee 1, who addressed a similar concern: "In the context of this paper, the IWC values are presented to give an impression on how the mass relation is between the single peaks. This is actually only possible if peaks were successfully separated first. Nevertheless, we decided to modify Fig 9 (former Fig. 6) in such a way that it now presents the total IWC (node 0) and the ratio of the IWC of the selected node to the total IWC."

P12L4: To my knowledge, microARSCL uses actually sub. I would recommend to focus stronger on the greater flexibility by overcoming the separation in noise separated and sub peaks.

We have removed the sentence on microARSCL and replaced it by (P16,L3ff): "The recursive structure of the tree allows to drop the artificial separation into noise-floor separated peaks and subpeaks within noise-floor separated peaks, as was necessary in prior approaches."

Appendix A: I would recommend adding a definition for spectral reflectivity.

We added a formula defining the spectral reflectivity (new Eq. A1).

P13L19: MIRA or Mira-35?

Changed to MIRA-35

---

## Author Response (AR3)

**General Remarks**

We thank Max Maahn and the second referee for the reading the manuscript again and making valuable suggestions. The referee comments are marked in grey and the response in black with indentation.

**Specific Reply to Referee #1**

I thank the authors for carefully addressing all my questions and comments. The manuscript has significantly improved and I have only a few specific comments and corrections to add before I can recommend it for publication.

>We thank the Referee for the positive opinion.

Minor:

P.3, L. 7: "The pulse repetition frequency was 5kHz and one Doppler spectrum was based on the fast Fourier transform of 256 pulses, yielding a Doppler resolution of 0.082ms^-1 (Tab. 1)." This sentence is maybe a bit misleading: You generate one raw Doppler spectrum based on 256 pulses. As you say a few sentences later, the actual spectrum stored and the one which is analyzed in your study is an average over 195 raw spectra. Thus, I suggest to write "raw spectra" in this sentence to make the difference to the final spectra more clear.

>We added 'raw' as suggested.

P.3, L.20: Why is the polarization leakage problem even more relevant for spectral LDR? Also later in L. 25: Are you requiring the spectrum in linear units to be a factor of 3 larger than the noise or does it relate to the dB space? Why factor of 3 and not 10? Is that based on previous studies or did you do your own sensitivity study using different thresholds and comparing the total LDR?

>Indeed, the initial sentence "Accurate measurements of polarization variables, like the LDR, depend strongly on instrument hardware due to polarization leakage." already sufficiently emphasizes the relevance of polarization leakage. We thus removed the sentence addressing the spectral LDR.
>The standard MIRA-35 spectra processing does the noise estimation only for the co channel and stores the spectral reflectivity for co and cross channel if in a bin signal in the co channel is above the noise. Hence for low SNR in the co channel LDR might artificially high in the cross channel. To prevent this we introduced this ad-hoc threshold guided on the factor 5 of Görsdorf et al [2017 JAOT]. However, we found the slightly lower factor of 3 to be sufficient for the rather long averaging time we use.

P.5, L.6: "A minimum is skipped, if the prominence of either of its subpeaks is less than 1dB" In Table 2 it says that "prom." has units of dBZ. Since it is defined as a difference, dB seems to be right.

>Corrected, thanks for pointing out.

P.5, L.10 (also P.7, L.4): "Reflectivity factor Z" should be Ze especially if you look at ice clouds.

>This point was already raised during the prior review and we inserted a respective section in Appendix A. To make this more clear also in the continuous text we modified the sentence to "Equivalent reflectivity factor Z (the subscript e is omitted for brevity) is calculated by integrating the spectral reflectivity of the whole peak (i.e. from the noise-floor up)".

Section 4.1: I think you should mention in the beginning of this section that this particular case has been extensively analyzed in Kalesse et al.2016 and that the reader can find there even more detailed information.

> Actually Kalesse et al 2016 [ACP] only discuss the 21 February 2014 case. The 02 February case is subject of Kalesse et al 2019 [AMT]. We have added a reference to this publication in the mentioned section.

Typos:
Abstract, L.8: "particle populations Arctic multi-layered" probably an "in" missing?
Abstract, L. 9: Programs' BAECC (?)
P.7, L.17: rule is based on (?)

> Corrected as suggested. Thanks!

**Specific Reply to Max Maahn**

I thank the authors for revising their manuscript, overall the quality of the paper has been significantly improved and I recommend the paper for publication subject to the following minor modifications:

> We thank the Referee for the positive opinion.

P2L9: Shouldn't this be multi-modal peaks?

> Thanks for pointing out. Corrected as suggested.

P4eq3: I don't understand the meaning of the brackets, isn't application of something like a floor function required to ensure you get the same parent for the left and right child?

> Yes, the brackets represent the floor function. We added a description after the equations.

P4L16: This is not correctly described if you have multiple peaks separated by noise. between '-v_nyq and +v_nyq' should be between the left edge of the left-most peak and the right edge of the right-most peak.

> Thanks for pointing to that misleading description. We omitted the part with v_nyq: "The root node contains all signal of the Doppler spectrum above the noise threshold."

P5L13: "node 0 is similar…' Also this statement is only true when there are not several, noise separated peaks in the Doppler spectrum. Many radar moment estimations (e.g. microasrcl) use the concept of the most significant peak. I.e. if there are two peaks in the radar Doppler spectrum that are separated by noise, the reflectivity corresponds only to the larger peak.

> Thanks for pointing us to this circumstance. We rephrased the passage accordingly:
> "Node 0 contains all components of the Doppler spectrum which are above the noise threshold. In general, this node is similar to the moment estimation commonly used to analyze Doppler spectra (e.g. Carter et al., 1995; Clothiaux et al., 2000; Görsdorf et al., 2015). Only in case of the presence of noise-separated subpeaks within node 0, some moment estimators such as microARSCL apply the moment retrieval to the most significant peak only. The child nodes (1-2) of node 0 […]"

Fig 3: In the text it sounds like the values would be normalized, but the number don't look like. Please add at least labels to avoid confusion.

Done as suggested.

P11L2: I mentioned that before but didn't express myself clearly enough: I still find it confusing that the authors distinguish the peak populations in the text by velocity ('faster-falling particle population', but Figure 8 separates by LDR. I recommend to make the naming consistent.

The selection is a hybrid of LDR, v and Z. The LDR was only used to guide the manual selection of the anchor nodes. The automated grouping algorithm then only relies on velocity and reflectivity. But we acknowledge, that the description is hard to follow and added more interconnections between fast falling and low LDR and slower falling and high LDR respectively.

**Minor changes**

P3L2: We removed the reference to Griesche et al. [2019 in preparation], as this publication is not submitted yet.

[revised manuscript text omitted]